# Identifying the impact of the covalent-bonded carbon matrix to FeN$_4$ sites for acidic oxygen reduction

Xueli Li[1] & Zhonghua Xiang[1✉]

The atomic configurations of FeN$_x$ moieties are the key to affect the activity of oxygen rection reaction (ORR). However, the traditional synthesis relying on high-temperature pyrolysis towards combining sources of Fe, N, and C often results in the plurality of local environments for the FeN$_x$ sites. Unveiling the effect of carbon matrix adjacent to FeN$_x$ sites towards ORR activity is important but still is a great challenge due to inevitable connection of diverse N as well as random defects. Here, we report a proof-of-concept study on the evaluation of covalent-bonded carbon environment connected to FeN$_4$ sites on their catalytic activity via pyrolysis-free approach. Basing on the closed $\pi$ conjugated phthalocyanine-based intrinsic covalent organic polymers (COPs) with well-designed structures, we directly synthesized a series of atomically dispersed Fe-N-C catalysts with various pure carbon environments connected to the same FeN$_4$ sites. Experiments combined with density functional theory demonstrates that the catalytic activities of these COPs materials appear a volcano plot with the increasement of delocalized $\pi$ electrons in their carbon matrix. The delocalized $\pi$ electrons changed anti-bonding d-state energy level of the single FeN$_4$ moieties, hence tailored the adsorption between active centers and oxygen intermediates and altered the rate-determining step.

[1] State Key Laboratory of Organic-Inorganic Composites, Beijing University of Chemical Technology, Beijing 100029, PR China.
✉email: xiangzh@mail.buct.edu.cn

The acute increase in the concentration of carbon dioxide ($CO_2$) in the atmosphere has intensified the urgent need for clean and sustainable energy technologies to achieve carbon neutralization as soon as possible[1]. Proton exchange membrane fuel cells (PEMFC) are typical clean energy conversion devices, whereas their sluggish oxygen reduction reaction (ORR) at the cathode requires highly efficient electrocatalysts to ensure the progress of the reaction[2,3]. Iron and nitrogen co-doped carbons (Fe–N–C) containing atomically dispersed $FeN_4$ moieties have been ever-increasing as the most promising alternative to commercialized platinum group metal due to maximized atom utilization and their desired activity[3–8]. It has exhibited catalytic activity approaching that of platinum in acids media as well as inspiring performance in PEMFCs[9–12].

Recently, some advanced characterization techniques such as Mössbauer spectra[13–15], X-ray absorption near-edge structure (XANES)[10,14,16] or annular dark-field scanning transmission electron microscope (ADF-STEM) analysis[8,11,17,18] have enabled identification for Fe–N–C atomic structure as well as determination toward its correlation with catalytic properties. Meanwhile, ab initio investigations have also deciphered the root of high instinct activity of $MN_xC_y$ moieties from the perspective of electron states based on a descriptor-based approach[19–21]. In general, the electrocatalyst activity is restricted to the number and intrinsic activity of active sites[12,22]. Studies manifest that the distinction between a high-loading and a low-loading electrocatalyst may be less than three orders of magnitude, however, the distinction in intrinsic activity between a superior electrocatalyst and an inferior electrocatalyst can overwhelmingly exceed ten orders of magnitude[23]. The importance of the intrinsic activity of the active site is self-evident. Unfortunately, the nature of the active site has remained elusive and various $FeN_xC_y$ sites were proposed as active centers. In particular, the $FeN_4C_{12}$, $FeN_4C_8$, or $FeN_4C_{10}$ moiety regularly functions as the typical active configuration in different studies[14,24–28], and thus frequently are employed as a benchmark model for active-site identification in the Mössbauer and XANES analysis.

Recently, Frédéric et al. have also demonstrated that their durability is inconsistent in the oxygen reduction process: the $FeN_4C_{12}$ moiety with a high-spin eventually inverted into iron oxides; whereas $FeN_4C_{10}$ moiety with low- or intermediate-spin remained unvaried after 50 h operation[13]. Moreover, Wu et al.[9,29] and Wang et al.[30] have together demonstrated that $FeN_4C_8$ and $FeN_4C_{12}$ moieties are more apt to absorb oxygen and proceed with four proton–electron transfer in oxygen reduction by electrochemical measurement coupled with density functional theory (DFT), compared to $FeN_4C_{10}$ moiety. These studies implicitly indicate the carbon matrix connected to $FeN_4$ sites has a non-negligible effect on the instinct activity of active sites. Nonetheless, the significance of carbon coordination environment connected to $FeN_4$ moieties has not been enough touched[31], and further in-depth studying still exists several critical issues: (1) The high-temperature pyrolysis of Fe, N, and C precursors hardly guarantee precise control of pure local carbon environment due to the inevitable connection with diverse N sometimes along with Fe; (2) the effect of randomly introducing defects anchored to the carbon plane is likewise inevitable.

Most recently, our groups have developed a pyrolysis-free strategy to prepare intrinsic conductive catalytic materials with well-designed $FeN_4$ centers, depending on the platform of covalent organic polymers (COPs)[32–35]. The tailorability of building blocks, as well as inherent durability indwelling in covalent bonds, endows them with excellent electrocatalytic characters[36–44]. Their well-defined configuration enables the oriented introduction of redox sites as well as electronegative heteroatoms into topological skeletons. Besides, the exclusive covalent linkages in COPs simultaneously endow their high chemical stability, which increases the feasibility of COPs application in ORR[45].

Herein, we report a proof-of-concept study on the evaluation of carbon environment covalent-connected to $FeN_4$ sites on their catalytic activity via this pyrolysis-free approach. Specifically, we designed a series of fully π-conjugated COPs, which contained well-defined $FeN_4$ configurations whereas discrepant delocalized conjugated carbon matrix connected to $FeN_4$ sites. The electrochemical tests demonstrated that the oxygen reduction activities of the four COPs exhibited a volcano plot with the expansion of the carbon skeletons adjacent to $FeN_4$ moieties in polymers. Further experiments preliminarily indicate the relationship between delocalized π-electrons (DDE) in carbon matrix adjacent to $FeN_4$ sites and electrocatalytic activity. The detailed DFT demonstrated that π-conjugated ligands connected to $FeN_4$ sites relocate electronic filling of antibonding states in Fe atom thereby modulating their electronic configurations and further altering their rate-determining step (RDS) in the ORR process. This study not only provides novel insights into the understanding ORR mechanism but also inspires people to seriously consider the effects of long-range electronic configuration in the carbon plane on active moieties.

## Results

**Synthesis of COPs with well-defined $FeN_4$ configuration and ascending DDE.** Instead of a high-temperature pyrolysis process randomly riveting Fe in the carbon matrix with inevitable connection with diverse N moieties, we adopted symmetrical acid anhydride or cyan groups as reactive groups to directionally prepare iron, nitrogen co-coordinated single-atom catalysts with well-designed configurations. Briefly, we deliberately assembled Tetracyanoethylene, Pyromellitic dianhydride, 1,4,5,8-Naphthalenetetracarboxylic dianhydride, and 3,4,9,10-Perylenetetracarboxylic dianhydride with Fe centers into extended $sp^2$ carbon networks by solid-phase synthesis, denoted by COP-Ene, COP-Ppcfe, COP-Nap, and COP-Pyr, respectively (Fig. 1, SEM images shown in Supplementary Fig. 1)[46,47]. The $^{13}C$ solid-state nuclear magnetic resonance (NMR) spectra (Supplementary Fig. 2) confirmed that four structures have been successfully synthesized, where C = N links connecting building blocks were smoothly arranged in corresponding networks, which was also favored by stretch vibration peaks of C = N covalent bonds at ~1600 $cm^{-1}$ in Fourier transform infrared spectroscopy (FT-IR spectra, Supplementary Fig. 3)[47,48]. More detailed characterization can be found in the Supplementary Materials.

Unlike conventional Fe–N–C catalysts, our pyrolysis-free strategy escapes unpredictable active-site configurations as well as accompanying diverse catalyst activities due to the pyrolysis process. The as-prepared COPs possess exclusive single Fe atom constrained in the conjugated 2D networks. Meanwhile, building blocks were orderly connected by rigid covalent bonds. As expected, Fe and N atoms kept a highly homogeneous distribution in the carbon matrix for four samples, which was demonstrated by the high dispersion image of Fe and N atoms in high-resolution transmission electron microscopy (HRTEM, Fig. 2a, b and Supplementary Figs. 4–8). Furthermore, we performed the aberration-corrected high-angle ADF-STEM test, which identified single Fe atoms from monodispersed bright spots images without metal clusters being observed (Fig. 2c and Supplementary Figs. 9 and 10) for all samples. To verify the coordination environments and chemical states of Fe atom in four samples, we carried out Fe K-edge analysis of X-ray absorption fine structure (XAFS). Synthetic four samples were demonstrated high similarity to the FePc benchmark, exhibiting

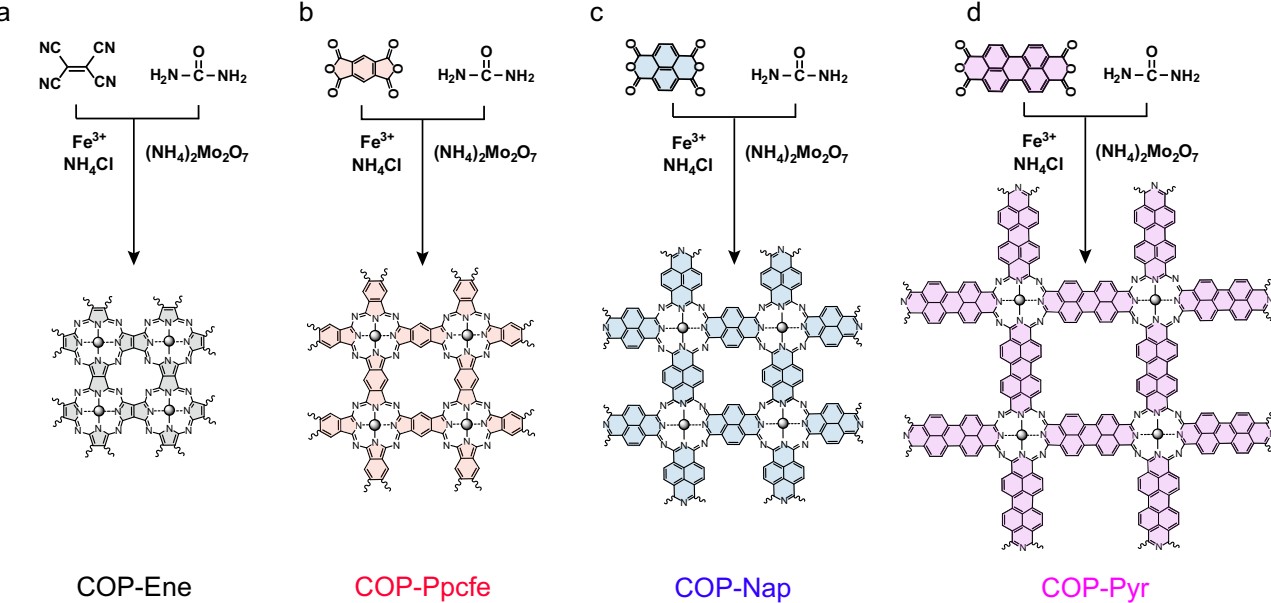

**Fig. 1 Synthesis routes of four full π-conjugated covalent organic polymers (COPs). a–d** Scheme of formation towards COP-Ene, COP-Ppcfe, COP-Nap, and COP-Pyr samples by tailoring monomers constructed corresponding building blocks.

dominated peaks corresponding to the Fe–N (~1.53 Å) and Fe–N–C (~2.7 Å) scattering paths, whereas the Fe–Fe bond (~2.18 Å) was not observed (Fig. 2d and Supplementary Figs. 11 and 12). Extended XAFS (EXAFS) fitting of four samples in *k*-space was also consistent with that of FePc benchmark and completely deviated from Fe foil, manifesting iron atom in synthetic samples existing as mononuclear centers (Fig. 2e and Supplementary Fig. 13). Besides, a predominant intensity maximum of wavelet transform (WT) contour plots of Fe K-edge in COP-Ppcfe sample appearing at approximately 4.0 Å$^{-1}$ in *k* space, which assigned to Fe–N(O) coordination (Fig. 2f), synergistically confirmed the above conclusions. EXAFS quantitative least-squares fitting analysis also supported that the constructed active centers were anastomotic with experimental EXAFS spectra of synthetic four samples (Supplementary Fig. 12 and Supplementary Table 1)[8,10,49].

Although these as-synthesized COPs present a semblable FeN$_4$ coordination structure, they have obviously a discrepant linked-carbon environment adjacent to FeN$_4$ sites. As investigated by FT-IR spectra (Supplementary Fig. 3), sequential ascending C=C stretching vibrations at ~1400 cm$^{-1}$ manifest successively incremental planar π-electron abundance in four polymers. In addition, continuously ascending deconvoluted π–π* peaks in C1*s* XPS spectra (Fig. 2g) consistently demonstrated that the DDE gradually increases with carbon skeleton covalent-connected FeN$_4$ sites in as-synthesized COPs. The typical sp$^2$ carbon conjugated peak at 284.3 eV in C1s XPS spectra is closely related to the transition of π-band (1*s* → π*)[50,51], therefore, full-width at half maxima value (FWHM) of conjugated C=C bond peaks can work as a criterion to quantificationally evaluate corresponding delocalization of π electrons in carbon matrix[52]. The narrower FWHM embodies fewer topological defects, higher DDE as well as stronger electron-donating capability in the carbon plane. As expected, observed significant shrink of FWHM along with the extension of *sp*$^2$ carbon-conjugated frameworks doubtlessly demonstrates that the DDE indwelled in polymers obviously increases (Fig. 2h). Based on the elemental analysis results of the C element, the gradually increasing C atom contents (C wt%, 33.18%, 47.53%, 56.65%, and 66.65% for COP-Ene, COP-Ppcfe, COP-Nap, and COP-Pyr, respectively) consistently reflects a

structural expansion in carbon skeletons. Besides, the gradual weakening of Fe$_{LMM}$ peak in related XPS spectra and decreasing metal iron contents obtained by ICP-OES with the augment of the carbon conjugated skeletons also suggest a stronger conjugated degree of the polymerization skeletons (Supplementary Figs. 14–16 and Supplementary Table 2). Room temperature $^{57}$Fe Mössbauer spectra further confirm that the effect of the DDE in the carbon matrix on the structure of FeN$_4$ moiety in four samples (Supplementary Fig. 17a and Supplementary Table 3). Generally, the structural varies of the active sites could be accurately reflected by the changes of quadrupole splitting (QS) value[53]. As shown in Supplementary Fig. 17b, a volcano plot appears between the structure of FeN$_4$ site in four samples (quantified by QS value of doublet FeN$_4$ moiety) and DDE (quantified by FWHM), reflecting that DDE in the carbon matrix induces the changes of the electronic configuration of FeN$_4$ moiety in four samples. It should be mentioned that the detection results of more than one doublet in the four samples and benchmark FePc may be attributed to extremely high sensitivity to energy changes on the order of 10$^{-8}$ eV for Mössbauer spectroscopy as well as the fact that FeN$_4$ moieties are greatly easy to adsorb oxygen[54]. Hereto, a series of FeN$_4$ sites with defined configurations yet increasingly extending surrounding carbon networks were design-oriented and systematically synthesized.

**Oxygen reduction performance and adsorption energy evaluation.** The ORR measurements for four COPs were carried out in 0.1 M HClO$_4$ to evaluate their performance discrepancy (Fig. 3a, Supplementary Fig. 18). Considering the limited conductivity of these materials, we added the carbon support (XC-72) in the electrochemical test to increase the conductivity of the catalyst and minimize the interaction between the carbon support and the COPs. The linear sweep voltammetry (LSV) curves revealed that the catalytic activity of the COP-Ppcfe sample was significantly ahead among the four COP samples, followed by COP-Ene; frustrating half-wave potentials mean that COP-Nap and COP-Pyr samples are difficult to undergo acidic ORR. The turnover frequency (TOF) and mass activity (Ma) per mass of metal at 0.7 and 0.8 V vs. RHE were further investigated to

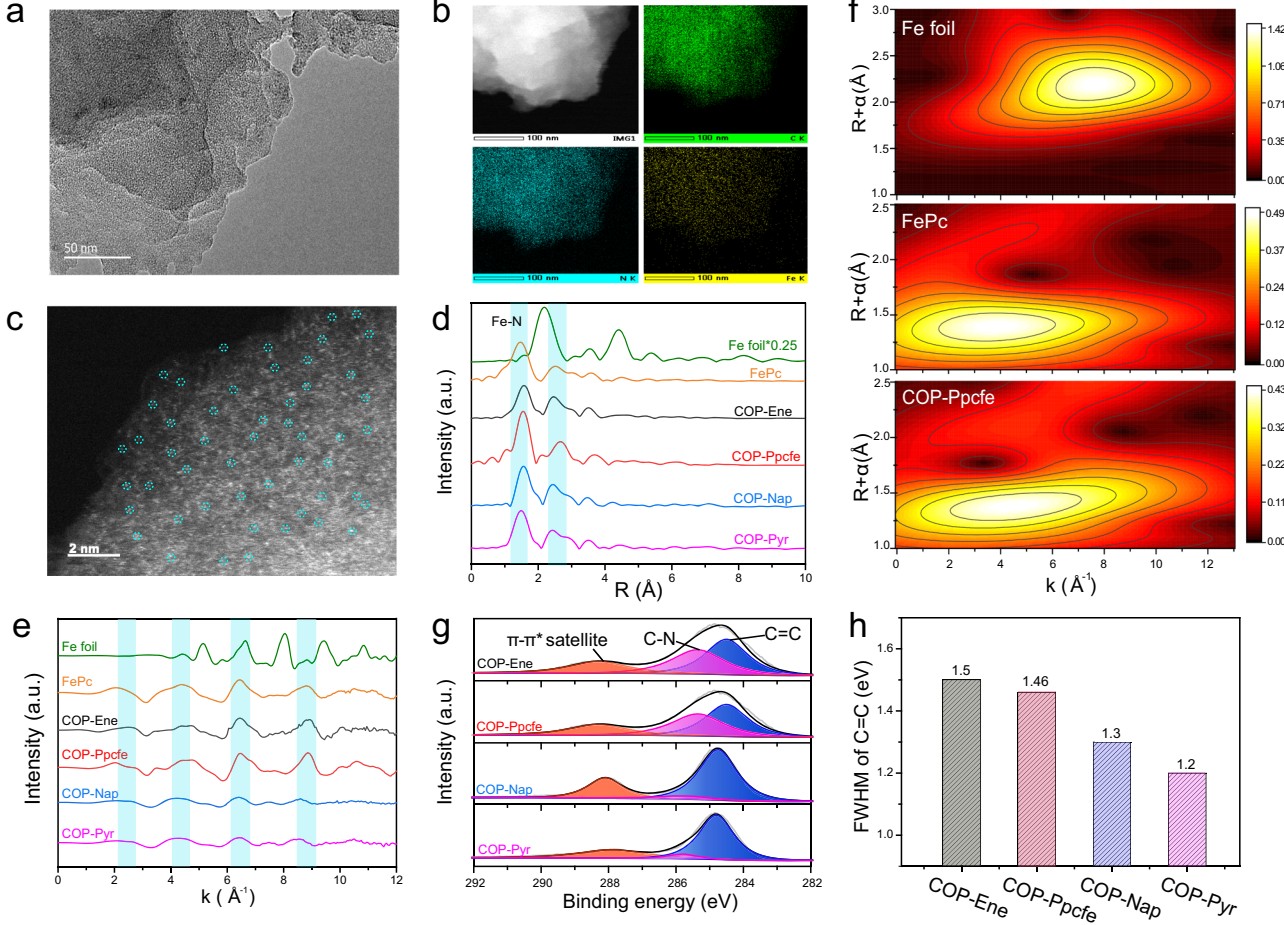

**Fig. 2 Morphology and structure characterization. a** The HRTEM images of COP-Ppcfe sample. **b** STEM image and the corresponding EELS elemental mapping of C (green), N (blue), and Fe (yellow) in COP-Ppcfe sample. **c** The HAADF STEM images of COP-Ppcfe sample. **d** The Fourier transforms of the EXAFS spectra for four samples, Fe foil, and FePc. **e** The corresponding EXAFS k space fitting curves of four samples, Fe foil, and FePc. **f** The Wavelet transform of COP-Ppcfe sample. **g** The C1s XPS spectra of the four as-synthesized COP samples. **h** The full-width at half maxima value (FWHM) of C = C bond in C 1s of the four as-synthesized COP samples.

compare the discrepancy in the electrochemical intrinsic activity of the active sites in the four as-synthesized COPs (Fig. 3b, c and Supplementary Figs. 19 and 20). As observed in Fig. 3b, both TOF and Ma exhibit a volcano plot that first increases and then decreases with the extension of sp² carbon skeletons connected to FeN₄ sites. In addition, the comparisons in kinetic current density ($J_k$), as well as the changes of Tafel slopes, also reflect the tendency of kinetic processes in ORR to become faster first and then slower with the increasement of DDE in the carbon matrix adjacent to FeN₄ moieties (Fig. 3c)[55]. The electrochemical experiments about pure COPs, COPs loaded with different carbon supports (such as soft carbon and rigid carbon) in acidic media as well as measurement in alkaline electrolyte also congruously exhibit similar volcano relationship (Supplementary Figs. 21–23). Aiming at these consistent volcano relationship plots, we next try to explain it from the adsorption strength of the reactant species obtained experimentally.

Theoretically, due to continuous multi-proton/-electron coupling reaction in the ORR process, large overpotential is ultimately stemmed from the mismatch of the adsorption energy[56]. Due to the restriction of scaling relations in oxygen intermediates, a volcano relationship[57] regularly emerges between ORR activity and adsorption energy, quantitatively illustrating Sabatier principle[19,20,58]. Namely, the binding between catalysts and oxygen intermediates is neither too strong nor too weak.

However, the direct measurement for adsorption strength in the experiment is a great challenge. Promisingly, this plight was eliminated by migrations of Fe²⁺/³⁺ redox potential ($E_{redox}$)[59]. As an effective indicator, the high $E_{redox}$ manifests weak Fe–O binding, and vice versa. As observed, a negative growth tendency in $E_{redox}$ appears with an order of COP-Ene, COP-Ppcfe, COP-Nap, and COP-Pyr in square wave voltammetry experiment (Supplementary Fig. 24), which indicates their growing adsorption strengths along with the extension of sp² carbon skeletons connected to FeN₄ sites.

Subsequently, we theoretically preliminarily identified the reliability of the above catalytic activity and adsorption strength. As observed, the theoretical $U^{on\text{-}set}$ potential by DFT calculations is consistent with experimental results (Fig. 3d). Especially, the negative $U^{on\text{-}set}$ potentials of COP-Nap and COP-Pyr samples imply that higher electrode potentials are required to overcome the grand barrier and afterward occur four-electron reaction. As a result, the sluggish desorption of OH* (OH* + H⁺ + e⁻ → H₂O) turns into RDS (vide infra, Supplementary Fig. 25), severely restricting the overall oxygen reduction kinetics. The experimental exchanged current densities ($j_0^{ept}$) directly scaled with theoretic $\Delta G_{max}^{-1}$ (Fig. 3e) also indicates transformation about the RDS[28]. Certainly, in addition to the four-electron reaction, O₂ also can be reduced to H₂O₂ through incomplete reduction. Therefore, the tropism of OOH* intermediates towards O*,

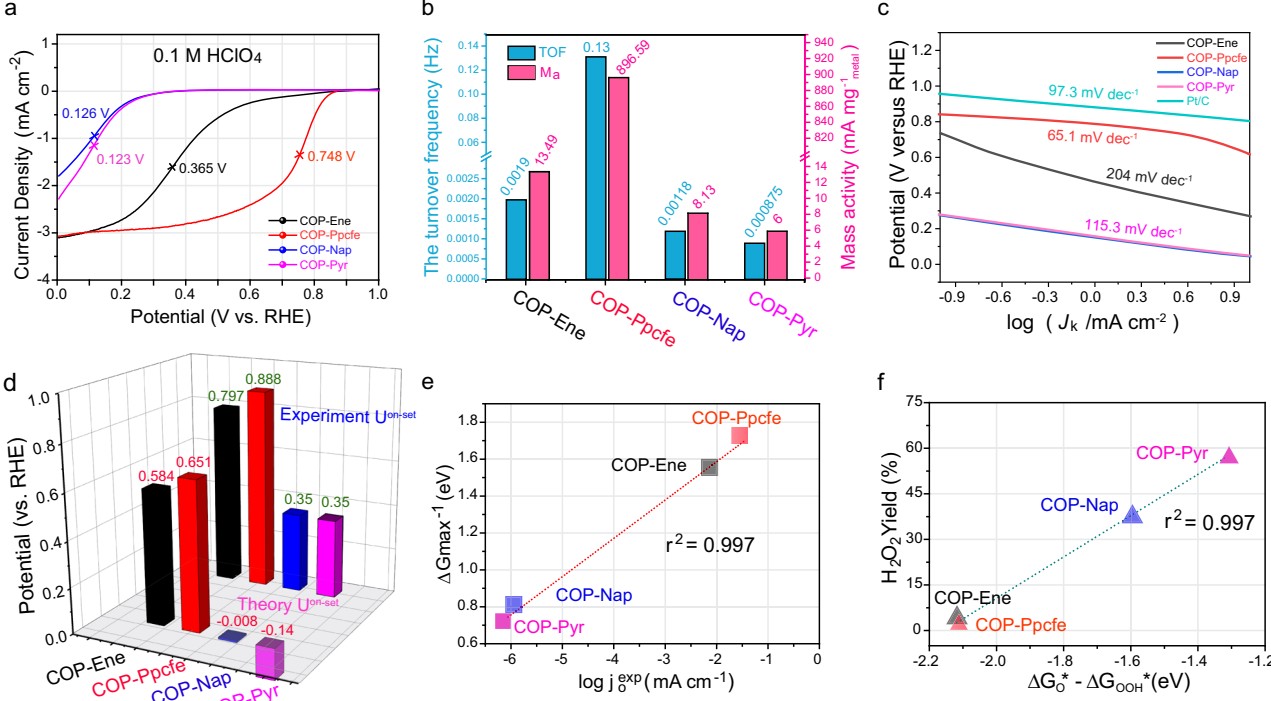

**Fig. 3 Electrochemical characterization. a** LSV curves of COP-Ene, COP-Ppcfe, COP-Nap, and COP-Pyr samples in $O_2$-saturated 0.1 $HClO_4$ solution at a scan of 5 mV s$^{-1}$ and a rotation speed of 1600 rpm. **b** The turnover frequency (TOF) at 0.7 V vs. RHE and mass activity (Ma) per mass of metal at 0.7 V vs. RHE. **c** Tafel plots. **d** Comparison of experimental and theoretical $U^{on\text{-}set}$ potential. **e** Linear relationship of theoretical reaction free energy of DRS ($\Delta G_{max}$) vs. the logarithm of exchange current density in the experiment ($j_0^{exp}$). **f** Linear relationship of experimental yields of $H_2O_2$ vs. theoretically thermodynamic propensity for O−O bind preservation ($\Delta G_{O^*}-\Delta G_{OOH^*}$).

described by $\Delta G_{O^*}-\Delta G_{OOH^*}$, is a criterion of the selectivity of $O_2$[60,61]. A more negative value indicates that oxygen is more facilely cleaved and completely reduced to $H_2O$. Therefore, as expected, a linear relationship between $\Delta G_{O^*}-\Delta G_{OOH^*}$ and $H_2O_2$ yields (obtained by rotation ring-disk electrode) can be found (Fig. 3f and Supplementary Fig. 26). Hereto, these results manifest that the oxygen reduction activities of the four COPs exhibit a volcano plot with the expansion of the carbon skeletons regardless in experiment or theory.

**The relationship between DDE and activity demonstrated by experiments.** Theoretically, the electron-donating capability of electrocatalysts determines the interfacial electronic transfer with oxygen intermediates, with an extensive effect on the oxygen reduction[17,62]. Generally, a catalyst with a small work function is more apt to undergo four proton–electron coupled steps, because electrons are effortless to escape from the catalyst surface[32]. According to the ultraviolet photoelectron spectroscopy (UPS) spectra (Supplementary Figs. 27 and 28), the obtained work function values (Ø) markedly decreased along with the increment of DDE in the carbon skeleton, thus $FeN_4$ sites with larger DDE are easier to react with oxygen[63]. Moreover, since the valence orbital is involved in the binding with associate oxygen intermediates, we further evaluated the valence band energy level, which is determined to be −1.37, −1.36, −1.27, and −1.02 eV relative to the Fermi level ($E_f$, Fig. 4a), respectively, for four COP-based catalysts. The obvious upshift of valence orbital reflects an inherent discrepancy of $d$-orbital level ($E_d$) along with DDE in the carbon matrix[64]. Therefore, the valence band level is employed as a criterion for evaluating the transition of the $d$-orbital level of the monometallic Fe atom in the synthesized COPs (vide infra).

In fact, the changes in adsorption are a direct cause of the ORR activity discrepancy. Therefore, we first associated oxygen adsorptions with DDE. As aforementioned, the DDE was obtained based on FWHM of C＝C bond (284.7 eV) while oxygen adsorption was quantified by $Fe^{2+/3+}$ redox potential ($E_{redox}$). In order to be clearer and more intuitive, we then employed $FWHM^{-1}$ and $E_{redox}^{-1}$ to express the intensity of DDE and adsorption strength: the larger their value, the stronger their strength. As observed from Fig. 4b, an increasing linear tendency between the $E_{redox}^{-1}$ and $FWHM^{-1}$ emerges, manifesting that the DDE in the carbon matrix connected to the $FeN_4$ sites is decisive to their adsorption strength. On the other hand, since $d$-orbital electronic states inherently determined the catalytic activity of $FeN_4$ sites, thereby the relationship between the valence band level and adsorption was next regressed and fitted (Fig. 4c). As a result, lifting valence band level ultimately induces stronger interactions between active sites and oxygen intermediates, which also corresponds to the drastically reduced work function values (Fig. 4a). Therefore, we have speculated that the DDE adjusts oxygen reduction activity largely by changing the valence-band or $d$-orbital electrons in active centers. As a result, although configurations of $FeN_4$ sites are similar, their electronic states are different and thus their intrinsic activities may also be diverse. Subsequently, we constructed the relationship between TOF and $FWHM^{-1}$. We noted that intrinsic catalytic activities in $FeN_4$ catalysts presented an approximate volcano plot with the increment of DDE near $FeN_4$ sites (Fig. 4d). The phenomenon is likely to be ascribed to that the relationship of DDE, $d$-orbital level, and adsorption are unidirectionally determined in sequence (Fig. 4b, c), while electrocatalytic activity and adsorption are restricted by Sabatier principle (Fig. 4e). Hereto, we speculate that the DDE in the carbon matrix connected to the $FeN_4$ sites affects the $d$-state level in the single iron atom, and thus tailors the

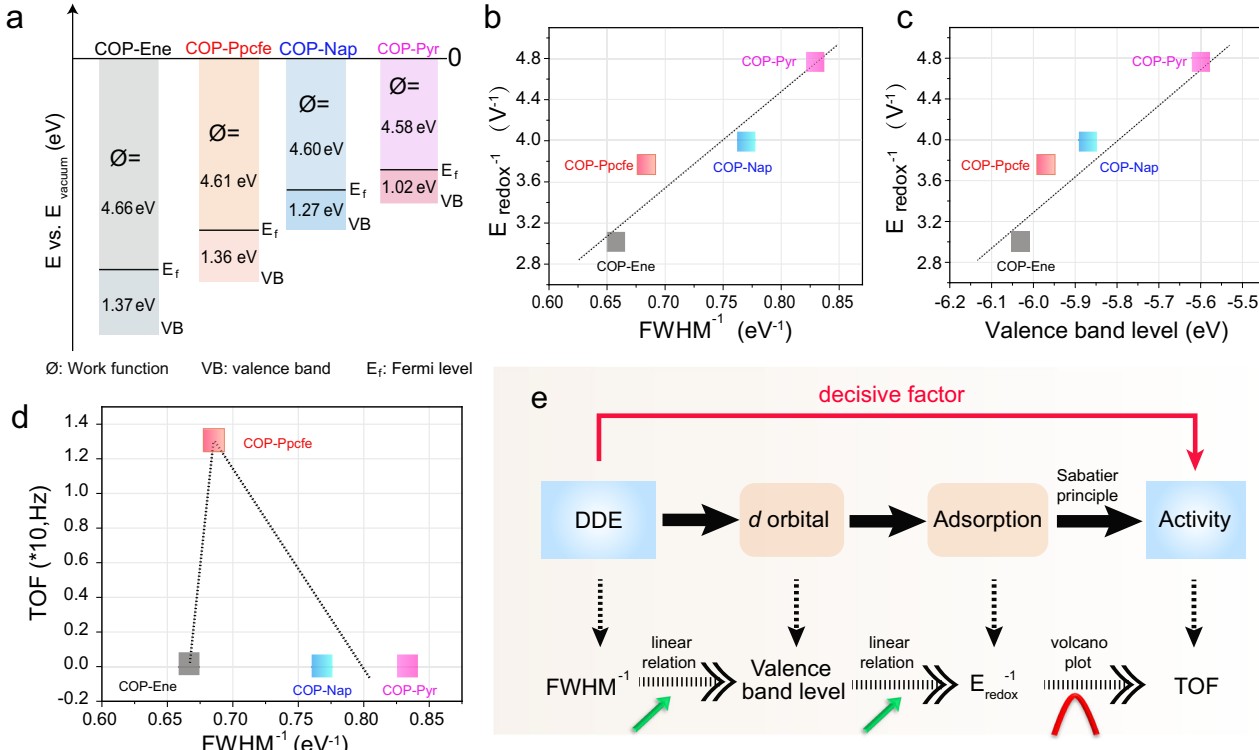

**Fig. 4 The relationship between DDE and catalytic activity. a** The schematic diagram of the valence band spectra and work function spectra (Ø) for COP-Ene, COP-Ppcfe, COP-Nap, and COP-Pyr samples ($E_f$ is the Fermi level). **b** The linear relationship of the $E_{redox}^{-1}$ vs. FWHM$^{-1}$. **c** The linear relationship of the $E_{redox}^{-1}$ vs. the valence band level. **d** Volcano plot between catalytic activity (quantified by TOF) and DDE (quantified by FWHM$^{-1}$). **e** Schematic diagram of the mechanism of DDE affecting catalytic activity.

adsorption strength between the moieties and oxygen intermediates; eventually, catalytic activity appears a volcano plot with DDE.

**Electronic structure analysis and mechanism study.** To comprehensively study the in-depth effect of the adjacent carbon environment on the ORR performance, we accomplished detailed DFT calculations. In addition to the four COPs discussed above, we further in silico designed two types of asymmetric carbon matrix structures connected to FeN$_4$ sites that are difficult to synthesize experimentally, named COP-Nap1 and COP-Nap2, to study the mechanism of DDE towards FeN$_4$ electronic structures (Fig. 5a). When we try to continue to increase the π conjugated moieties, the models of macrocyclic structure exhibit a certain degree of skeleton bending, indicating that further incremental conjugated structures will reduce the stability of polymers (Supplementary Fig. 29). In six models, there is no obvious relationship both in Fe–Fe bond length in adjacent building blocks and Fe–N bond length, which firstly excludes the effects of single atom density or nitrogen atom types on ORR performance (Supplementary Figs. 30 and 31). The climbing total electron numbers observed from the total density of states (Supplementary Fig. 32) and electronic localization functions diagrams (Supplementary Fig. 33) theoretically demonstrated the DDE in carbon matrix is gradually increasing with the expansion of the carbon skeleton connected to FeN$_4$ sites, which also inosculates with the FWHM of conjugated C = C bond peaks experimentally (Fig. 2g, h). The Bader charge population analysis manifests that these carbon matrix functions as a "motor", gradually donating more electrons from themselves into FeN$_4$ sites thereby altering the net charge of the single Fe atom (Fig. 5b, Supplementary Table 4), strengthening iron oxophilicity and binding with adsorbates[65–67].

Generally, for most Fe–N–C catalysts, protonation of $O_2^*$ ($O_2^* + H^+ + e^- \rightarrow OOH^*$) or desorption of OH* (OH* + H$^+$ + e$^- \rightarrow H_2O$) is RDS in oxygen reduction[68]. The protonation of $O_2^*$ is the initial step and directly affects the next electron transfer. Therefore, we initially focused on the changes in the electron states of oxygen adsorption to unveil the effect of the DDE on oxygen reduction. Among six systems, COP-Ene, COP-Ppcfe, COP-Nap1, and COP-Nap2 display end-on adsorption models, whereas COP-Nap and COP-Pyr exhibit side-on adsorption models (Fig. 5c). For FeN$_4$ sites with abundant DDE, such as COP-Nap and COP-Pyr samples, the Fe atom occupied orbitals transfer more electrons into anti-bonding orbitals of $O_2$ via two O atoms, thereby adsorbed dioxygen adopts side-on configuration on Fe atom (transferred electrons ~0.65|e| in side-on configuration vs. 0.3–0.4|e| in end-on configuration). Partial density of state analysis (Fig. 5d) indicates that electrons mainly transfer from $3d_{xz}$ and $3d_{yz}$ orbitals in Fe atom to $1\pi^*$ orbitals in dioxygen through π-back bonding. In contrast, for other COPs, superoxide species form when $O_2$ molecules are absorbed on the Fe atom by end-on interaction, where electrons mainly transfer from the Fe $3dz^2$ orbitals to the oxygen $1\pi^*$ orbitals and form σ bonds. These completely antithetical manners imply that even if active sites possess semblable geometric structures, their electronic transfer paths and bonding patterns may be completely different (Fig. 5d and Supplementary Figs. 34 and 35)[69].

Distinguish from the interaction between Fe atoms and $O_2^*$, the $1\pi$ valence orbital in lone pair O $2p_x$, $2p_y$ electrons, and $3\sigma$ orbital in H 1s–O $2p_z$ of OH* intermediate occurred renormalizing when OH* reacted on $3d$ orbitals of Fe atom (Fig. 5e and Supplementary Fig. 36)[70]. This coupling makes their energy level split into bonding states under the Fermi level and antibonding

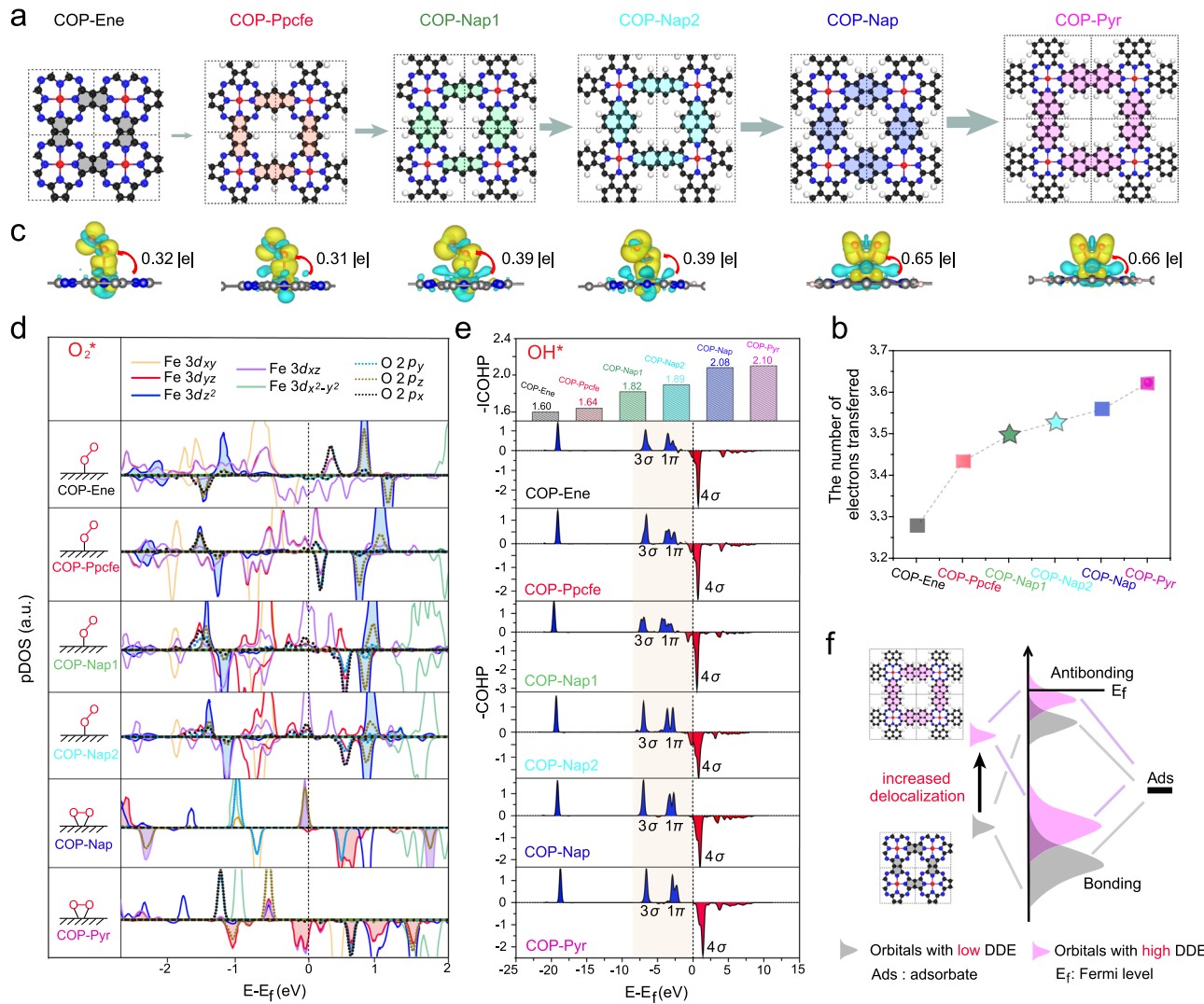

**Fig. 5 The ORR mechanism revealed in depth through DFT. a** Six models of COPs architectures: COP-Ene, COP-Ppcfe, COP-Nap1, COP-Nap2, COP-Nap, COP-Pyr. (In addition to the four COPs synthesized experimentally, we have also constructed Nap1 and Nap2 models to enrich the DDE in the carbon matrix. The red, dark blue, white, and black colors represent Fe, N, H, and C atoms, respectively. To distinguish their structure discrepancy, the carbon skeleton connecting the two FeN$_4$ sites is displayed in different colors on purpose). **b** Taking FeN$_4$ as a whole, the numbers of electrons are transferred from the surrounding C skeleton to the FeN$_4$ moieties (constructed COP-Nap1 and COP-Nap2 models are marked by the star symbol). **c** The most stable configurations for oxygen adsorption and charge density differences of O$_2$ chemisorbed on different Fe–N–C moieties (cyan and yellow represent positive and negative charges, respectively). **d** Projected density of states (pDOS) plots of 2p orbitals (2p$_x$, 2p$_y$, 2p$_z$) for O$_2$ and 3d orbitals (3d$_{xy}$, 3d$_{yz}$, 3d$_z{}^2$, 3d$_{xz}$, 3d$_{x^2-y^2}$) for corresponding COPs. **e** Crystal orbital Hamilton population (COHP) analysis and corresponding Integrated crystal orbital Hamilton population (ICOHP) value of OH* absorbed on corresponding COP surfaces. **f** Schematic diagram of orbital hybridization of the adsorbates bonding orbitals and 3d orbitals of COPs with the increment of DDE.

states above the Fermi level (Fig. 5f)[71]. The higher the energy of $E_d$ relative to $E_f$ means the less electronic occupancy in the antibonding states and corresponding stronger adsorption[19]. This precise tailor pointing at $E_d$ energy level was also validated by a series of increasing absolute values of integrated crystal orbital Hamilton population value (ICOHP, Fig. 5e)[72]. Thus, it can be concluded that although the $d$-orbitals electron in FeN$_4$ sites determines the progress of ORR by affecting oxygen intermediates, π-conjugated ligand configurations connected to FeN$_4$ sites can relocate electronic filling of antibonding states in Fe atom and modulate electronic configurations of FeN$_4$ sites.

Finally, we studied ORR pathways and catalytic activities of the six well-defined catalytic models in the oxygen reduction process. Although four-electron/ proton coupling reaction theoretically includes the associative mechanism and the dissociative mechanism, the grand barriers of directly breaking the O=O bond

means that the dissociative mechanism for uniform and continuous single-atom catalysts is insurmountable (Supplementary Fig. 25). Nevertheless, the delocalized π electrons in the carbon matrix can change the RDS of the four-electron ORR (Fig. 6a, b, Supplementary Fig. 37, Supplementary Tables 5–7). As we all know, the intrinsic activity of the electrocatalysts can be determined by the limiting reaction barrier in the RDS. For COP-Ene, COP-Ppcfe, COP-Nap1, the RDS is oxygen protonation to OOH* with limiting barriers of 0.64, 0.58, and 0.56 eV, respectively (Fig. 6c). However, when the degree of delocalization in the carbon matrix continues increasing, the RDE of ORR was transformed from protonation of O$_2$* into desorption of OH*, with a limiting barrier of 0.78 eV for COP-Nap2, even as large as 1.24 and 1.37 eV for COP-Nap and COP-Pyr. In addition, the free energy of the six COP models is more negative, successively suggesting that the DDE in the carbon matrix induces stronger

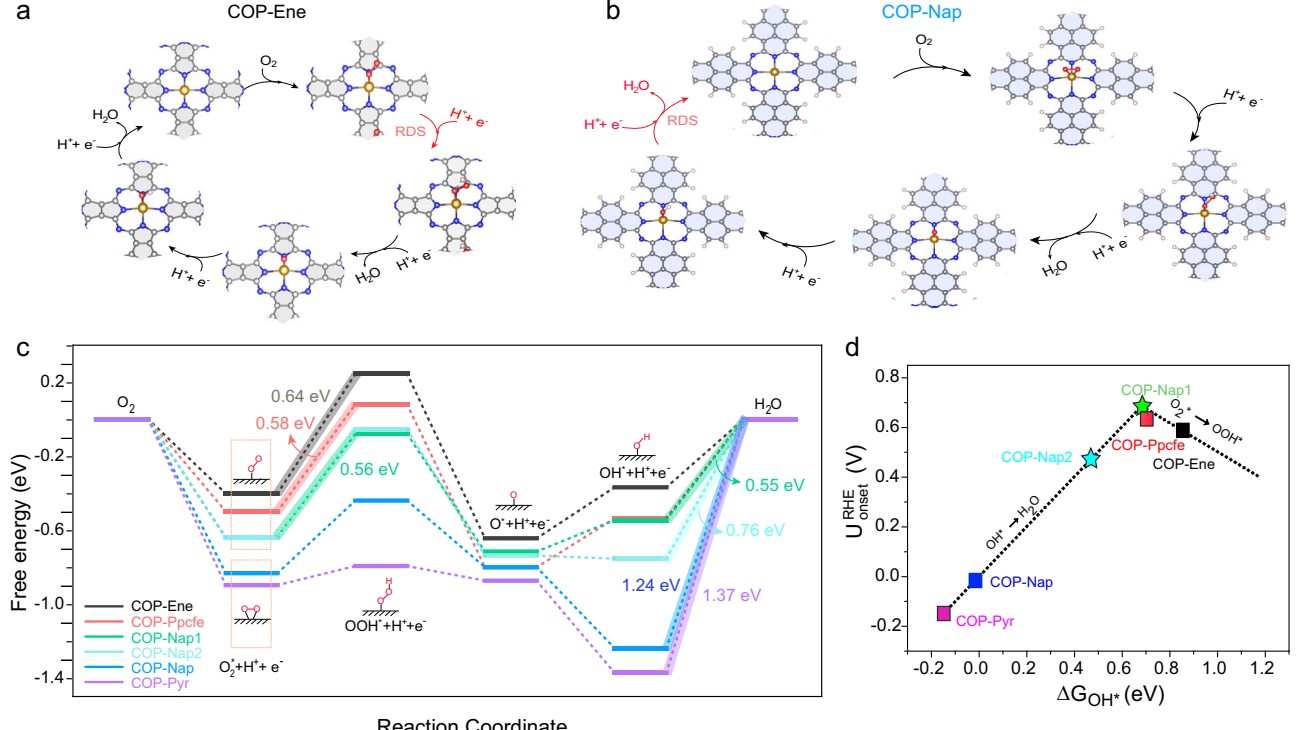

**Fig. 6 Schematic diagram of rate-determining step transition. a, b** Proposed ORR reaction scheme with the intermediates towards COP-Ene (**a**) and COP-Nap (**b**). RDS is a rate-determining step. The blue, gray, red, pink, and golden spheres represent N, C, O, H, and Fe, respectively. **c** The Free energy diagram of COP-Ene, COP-Ppcfe, COP-Nap1, COP-Nap2, COP-Nap, and COP-Pyr at equilibrium electrode potential. **d** The volcano plot of onset potential versus adsorption energy of OH*.

chemical adsorption with oxygen intermediates, which is also the direct reason for the transition of RDS. By correlating the onset potential and Gibbs free energy of OH* intermediates, a typical volcano plot was constructed. Notedly, the "JUST" DDE in carbon matrix connected to FeN₄ sites makes COP-Ppcfe, and COP-Nap1 have the optimum *d*-state level, thereby exhibit 'RIGHT' adsorption, lie at the apex of volcano plot and exceed most of the same type of pyrolysis-free catalysts in the experiment (Fig. 6d, Supplementary Table 8). Therefore, our calculation results manifest that the carbon matrix connected to FeN₄ sites remarkably contributes much to the catalytic activity of electrocatalysts and the RDS in oxygen reduction processes.

## Discussion

In summary, we have constructed a series of fully π-conjugated COPs with analogous FeN₄ configurations to study-orient the effect of carbon environment covalent-connected to FeN₄ sites on their catalytic activity. Significantly, the FWHM, valence-band spectra, $Fe^{2+/3+}$ redox potential, as well as electrochemical measurements together reveal that the relationship of DDE, *d*-orbital level, and adsorption is unidirectionally determined in sequence, while electrocatalytic activity and adsorption are restricted by Sabatier principle, exhibiting a volcano plot. The DFT calculations further analyze that those superior intrinsic active moieties are stemmed from the moderate electron filling of the d-state antibonding orbitals of Fe center in the FeN₄ sites, which is directly related to the π-conjugated electrons in the connected carbon matrix. This study not only clarified the origin of intrinsic-activity diversity of FeN₄ catalysts but also provided a strategy for regulating the oxygen reduction performance of the active site by changing the electronic configuration of the carbon atom covalent-connected to the FeN₄ sites.

## Methods

**Pyrolysis-free approach to synthesize electrocatalytic materials. Preparation of COP-Ene sample**. In brief, for the synthesis of metal COP-Ene sample, tetra-cyanoethylene (0.320 g), urea (2.4 g), NH₄Cl (0.2675 g), and (NH₄)₆Mo₇O₂₄·4H₂O (0.08 g) and FeCl₃ (0.203 g) were mixed and ground adequately; the mixture was then transferred into the crucible, and heated in a muffle furnace at 180 °C for 5 h. After cooling to room temperature, the product was washed repeatedly with 1 M HCl, 0.5 M NaOH, and ethanol by soxhlet extraction. The precipitates were dried under vacuum at 60 °C overnight.

**Preparation of COP-Ppcfe sample**. In brief, for the synthesis of metal COP-Ppcfe sample, pyromellitic dianhydride (PMDA, 0.525 g), urea (2.4 g), NH₄Cl (0.2675 g), (NH₄)₆Mo₇O₂₄·4H₂O (0.155 g) and FeCl₃ (0.225 g) were mixed and ground adequately; the mixture was then transferred into the crucible, and heated in a muffle furnace at 220 °C for 3 h. After cooling to room temperature, the product was washed repeatedly with 1 M HCl, 0.5 M NaOH, and ethanol by soxhlet extraction. The precipitates were dried under vacuum at 60 °C overnight to obtain COP-Ppcfe.

**Preparation of COP-Nap sample**. In brief, for the synthesis of metal COP-Nap sample, 1,4,5,8-Naphthalenetetracarboxylic dianhydride (0.67 g), urea (2.4 g), NH₄Cl (0.2675 g), and (NH₄)₆Mo₇O₂₄·4H₂O (0.155 g) and FeCl₃ (0.225 g) were mixed and ground adequately; the mixture was then transferred into the crucible, and heated in a muffle furnace at 275 °C for 3 h. After cooling to room temperature, the product was washed repeatedly with 1 M HCl, 0.5 M NaOH, and ethanol by soxhlet extraction. The precipitates were dried under vacuum at 60 °C overnight.

**Preparation of COP-Pyr sample**. In brief, for the synthesis of metal COP-Pyr sample, 3,4,9,10-Perylenetetracarboxylic dianhydride (0.98 g), urea (2.4 g), NH₄Cl (0.2675 g), and (NH₄)₆Mo₇O₂₄·4H₂O (0.155 g) and FeCl₃ (0.225 g) were mixed and ground adequately. The mixture was then transferred into the crucible, and heated in a muffle furnace at 300 °C for 3 h. After cooling to room temperature, the product was washed repeatedly with 1 M HCl, 0.5 M NaOH, and ethanol by soxhlet extraction. The precipitates were dried under vacuum at 60 °C overnight.

**Electrochemical measurements**. All electrochemical measurements were carried out at room temperature by using a CHI760E electrochemical analyzer with a typical three-electrode electrochemical cell. In these cases, saturated calomel electrode (SCE), carbon rod, and glass carbon rotating disk electrode coated with catalysts were used as reference electrode, a counter electrode, and the working

electrode, respectively. Electrode potentials were converted to reversible hydrogen electrode (RHE), using the following relationship (ERHE = ESCE + 0.241 + 0.059 pH).

**Preparation of catalyst ink**. 5 mg as-synthesis catalysts were dispersed in a 1000 μL mixture solution containing ethanol (950 μL) and Nafion solution (0.5 wt%, 50 μL). Another 5 mg XC-72 carbon black was added to the ink to improve the conductivity of the catalyst. The ink was then shaken and sonicated in a bath sonicator for at least 30 min to form a homogeneous suspension. Then 10 μL suspension (containing 50 μg of catalyst) was loaded onto a glass carbon electrode surface with a diameter of 5 mm and resulted in a mass loading of about 0.255 mg cm$^{-2}$. Cyclic voltammetry (CV) measurements were carried out at potential from 0 to 1.1 V (vs. RHE) at a scan rate of 100 mV s$^{-1}$ and LSV measurements were performed with a scan rate of 5 mV s$^{-1}$ at a rotating rate of 1600 rpm. The electrolyte (0.1 M HClO$_4$) was bubbled by high-purity O$_2$ or N$_2$ for 30 min into saturation state before each test.

The rotating ring-disk electrode (RRDE) examinations were performed with the Pt ring electrode (the potential of Pt ring was set at V = 1.1 VRHE) to test the ring current (Iring). The polarization curves were examined at a disk rotation rate of 1600 rpm and catalyst loading was 0.255 mg cm$^{-2}$. The peroxide yield (H$_2$O$_2$%) and the electron transfer number ($n$) were calculated by: $n = 4 \times I_d/(I_d + I_r/N)$ and H$_2$O$_2$% = $200 \times (I_r/N)/(I_d + I_r/N)$, where $I_d$ represents the disk current and $I_r$ represents the ring current, $N$ represents the RRDE collection efficiency, about ~0.37 in our system.

The TOF of COP-Ene, COP-Ppcfe, COP-Nap, and COP-Pyr were calculated by: TOF = $(J_k \times s)/(4 \times m \times F)$, where $J_k$ represents kinetic current density at 0.7 or 0.8 V, $s$ represents the surface area of the glass carbon electrode, $m$ represents the mole numbers of FeN$_4$ active moieties on the electrode, $F$ is the Faraday constant (96,500 C mol$^{-1}$). The Ma per mass of metal was calculated by Ma = $J_k/m$, where $J_k$ represents kinetic current density at 0.7 V, m represents the contents of metal in the glass carbon electrode.

**Square wave voltammetry experiment**. Square wave voltammetry test was performed in N$_2$-saturated 0.1 M HClO$_4$ electrolyte with a step potential of 1 mV, amplitude of 1 mV, and scan frequency of 10 Hz.

**Characterizations**. FE-SEM (JSM-6701/JEOL) was used to observe the morphologic and structural characteristics of the samples. Monodisperse morphology and further single atom character of the sample were obtained by HRTEM (JEM 2200FS) and the high-angle annular dark-field scanning transmission electron microscopy (HAADF-STEM, JEM-ARM300F). The FT-IR analysis was performed on a Nicolet 8700/Continuum XL with wavenumber from 2200 to 500 cm$^{-1}$. Solid-state NMR spectra were measured on a Burker 400 M spectrometer operating at 10 kHz for $^{13}$C. Chemical compositions and elemental oxidation states of the samples were investigated by X-ray photoelectron spectroscopy (XPS, Thermo Fischer, ESCALAB). The elemental spectra were all corrected concerning C1s peaks at 284.8 eV. UPS (PHI5000 VersaProbe III, Scanning ESCA Microprobe with SCA Spherical Analyzer) was collected using He I (21.2 eV) radiation. The content of metallic iron in the four samples was gained by an inductively coupled plasma Optical Emission Spectrometer (ICP-OES, Agilent 7700X & Agilent 7800). The X-ray absorption spectra including XANES and EXAFS of the samples at Fe K-edge (6974–8110 eV) were collected at Beijing Synchrotron Radiation Facility (BSRF), where a pair of channel-cut Si (111) crystals was used in the monochromator. The Fe K-edge XANES data were recorded in a transmission mode. Fe foil, Fe$_2$O$_3$, and FePc were used as references. The storage ring was working at the energy of 2.5 GeV with an average electron current of below 200 mA. The acquired EXAFS data were extracted and processed according to the standard procedures using the ATHENA module implemented in the IFEFFIT software packages. The obtained XAFS data was processed in Athena (version 0.9.26) for background, pre-edge line, and post-edge line calibrations. Then Fourier transformed fitting was carried out in Artemis (version 0.9.26). The $k^3$ weighting, $k$-range of 3–12 Å$^{-1}$, and $R$ range of 1-~3 Å were used for the fitting. The four parameters, coordination number, bond length, Debye–Waller factor, and $E_0$ shift (CN, $R$, $\sigma^2$, $\Delta E_0$) were fitted without anyone being fixed, constrained, or correlated. For WT analysis, the $\chi(k)$ exported from Athena was imported into the Hama Fortran code. The parameters were listed as follows: $R$ range, 1–4 Å, $k$ range, 0–13 Å$^{-1}$; $k$ weight, 2; and Morlet function with $\kappa = 10$, $\sigma = 1$ was used as the mother wavelet to provide the overall distribution. Mössbauer measurements were performed using a conventional constant acceleration type spectrometer in transmission geometry in the temperature range from 300 to 30 K. Absorbers were prepared in powder form (10 mg of natural Fe cm$^{-2}$). The ɤ-ray source is powder is a commercial 25 mCi $^{57}$Co in sodium nitroprusside powder and all isomer shifts were quoted relative to the α-Fe foil at room temperature.

**Computational methods**. We employed the spin-polarized DFT method for all calculations, as implemented in the Vienna ab initio Simulation Package code. The projector-augmented-wave basis set was adopted to describe ion-electron interaction with cut-off energy of 500 eV. Perdew–Burke–Ernzerhof of the generalized gradient approximation functional was used to treat exchange-correlation

interaction. The Γ center 4*4*1 k-points grids were chosen for optimization of COP-Ene, and 3*3*1 k-points grids were chosen for optimization of COP-Ppcfe, COP-Nap1, COP-Nap2, COP-Nap, and COP-Pyr. The denser Γ center 9*9*1 k-points grids was chosen for electronic structure calculations. The convergence tolerance of energy and force was 10$^{-5}$ eV and 0.01 eV/Å. A 15 Å thickness vacuum was inserted to eliminate the interaction induced by a periodic boundary condition. The Gibbs free energy ($\Delta G$) diagram of the ORR was calculated using the computational hydrogen electrode model proposed by Nørskov et al, where the free energy of (H$^+$ + e$^-$) under standard conditions is equal to the value of 1/2 H$_2$. Free energy is calculated by the formula $\Delta G = \Delta E + \Delta ZPE - \Delta TS + \Delta G_U + \Delta G_{PH}$, where $E$ is the total energy of the system, ZPE is the zero-point energy, and S is the entropy. ZPE corrections were calculated as ZPE = $1/2hv_i$, where h is Planck's constant and vi is the frequency of the vibrational mode of binding molecules. Since DFT cannot accurately obtain the Gibbs free energy of oxygen molecules, this value can be obtained indirectly through water and hydrogen ($G_{O2} = 2G_{H2O} - 2 G_{H2} + 4.92$ eV). The free energy of liquid water is equal to the free energy of gaseous water at 0.035 atm and 298.15 K. The oxygen-containing intermediates O* and OH* are stabilized by the solvation effect of ~0.3 eV[19]. In acidic media, the pH is assumed to be 0. The electrode potential adopts the reversible hydrogen electrode potential.

## Data availability
The data that support the findings of this study are available from the corresponding author on reasonable request.

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

## Acknowledgements

This work was supported by the National Key Research and Development Program of China (2017YFA0206500); the NSF of China (21922802); the Beijing Natural Science Foundation (JQ19007); Talent cultivation of State Key Laboratory of Organic-Inorganic Composites; "Double-First-Class" construction projects (XK180301 and XK1804-02); distinguished Scientist Program at BUCT (buctylkxj02). We also thank the 1W1B station for XAFS measurements in Beijing Synchrotron Radiation Facility (BSRF).

## Author contributions

X.L. performed the synthesis, structural characterizations, electrochemical tests, and DFT calculations. X.L. and Z.X. wrote this paper. Z.X. supervised and led this project. All authors provided critical feedback and helped shape the research and paper. All authors commented on the paper. We also thank Prof. Pinxian Xi and Shixin Hu from Lanzhou University (Lanzhou, China) for the measurements and discussion of Mössbauer spectra.

## Competing interests

The authors declare no competing interests.
