## [Peer review file · Nature Communications]

REVIEWER COMMENTS

Reviewer #1 (Remarks to the Author):

This paper investigated the impact of carbon moiety adjacent Fe-N_x sites in Fe-N-C catalysts. The authors combined a bottom-up synthesis with low-temperature (220 °C) pyrolysis to afford four model Fe-N-C catalysts having carbon moieties with different degree of pi conjugation. COF-Ppcfe showed the best activity, and the samples exhibited a volcano-type relationship of ORR activity and degree of pi conjugation. Some aspects of these results may be of interest in the field, and the model catalyst approach is intriguing. However, many important claims were not fully substantiated by experimental evidence. Also, the manuscript contains too many typos and awkward expression, making hard to follow the manuscript. Overall, this review cannot support the publication of this work in Nature Comm.

1. Fe contents in all catalysts should be quantified by ICP-OES analysis. Also, Mo that was used as the catalyst for generating COPs can remain after the polymerization, which should also be checked with ICP-OES analysis.
2. The samples other than COP-Ppcfe should be fully characterized using STEM and EXAFS to support that all samples exclusively have Fe-N sites.
3. Although the ORR activity of COP-Ppcfe was the best among the samples and its half-wave potential (0.748 V vs RHE) is impressive, its diffusion current density (3 mA/cm²) is too low. According to the Levich equation with complete 4-electron selectivity, diffusion current density should be 6 mA/cm² at a rotation speed of 1600 rpm. Most of highly active Fe-N-C catalysts show diffusion current density of 5.5-6 mA/cm².
4. In relation to the above question, what is four-electron selectivity of the catalysts?
5. Other catalysts showed very low catalytic activity? If they truly contain Fe-N sites, their activities are unusually low.
6. The manuscript has too many awkward expression and typo. Some examples include
 - 1) Abstract: Experiments coupled with density functional theory demonstrate that the catalytic activities appear a volcano plot with the increase of degree of delocalized π electrons from the carbon matrix.
 - 2) Line 81-83: The electrochemical tests demonstrated that the oxygen reduction activities of the four COPs exhibited a volcano plot with the expansion of the carbon skeletons.
 - 3) Line 85-89: The detailed DFT demonstrated that π -conjugated ligand configurations connected to FeN₄ sites relocate electronic filling of antibonding states in Fe atom and modulate their electronic configurations, hence altered the rate-determining step (RDS) in ORR process.
 - 4) Title: Identify 0 Identifying
 - 5) Line 29: acutely 0 acute
 - 6) Line 34-35: "Iron and nitrogen co-doped carbons (Fe-N-C) electrocatalysts"
 - 7) Line 54-55: "In particular, the FeN₄C₁₂, FeN₄C₈, and FeN₄C₁₀ moiety regularly separately function as typical active configuration..."
 - 8) Line 97: "prepare-oriented"
 - 9) Line 123: "as-synthesis"
 - 10) Line 158: "harsh" This experimental condition is not harsh.

Reviewer #2 (Remarks to the Author):

This manuscript reports the impact of covalent-bonded carbon matrix to the so-called active site Fe-N_x by a pyrolysis-free method. Embedded Fe-N₄ into covalent organic polymer (COP) is an effective approach to keep the FeN₄ unit unchanged during the synthesis, unlike the pyrolysis process. This synthesis can investigate the configuration of FeN₄ exclusively on the ORR activity. They synthesize 4 COPs with various strengths, defined as DDEs, for the study of ORR activity and give a volcano plot.

They use a series of methods to characterize the materials and also use DFT calculations as well, trying to unveil the intrinsic mechanism of ORR under the interaction between the FeN₄ and covalent-bond matrix. The manuscript is well written and the results are of interest. I would suggest its publication in Nat Comm by some revisions.

1. It is better if the authors can synthesize more COPs for the volcano plot.
2. Besides activity, stability is also important for a catalyst. So it is better to give some investigation on this point.
3. COPs as a model catalyst for mechanism study is fine, but the number of active sites in COPs should also be discussed as dense active sites are necessary for an efficient catalyst.
4. In the Introduction and Discussion, the recent reported asymmetrically electronic distribution of an active site plays important roles on the activity. This may be helpful for better understanding of the activity dependence on the FeN_x/matrix interaction through electron transfer (directly influences the electronic asymmetry). This discussion can be referred to Adv Mater 2019, 1805581; and 2020, 32 (16), 2000966.
5. It should also discuss the potential design of active sites and assembly as a catalyst, which may be the topic of research for next generation catalysts. COPs or COFs (JACS 2020, 142, 8104) could be the effective approaches but require more efforts to construct catalysts with high-dense active sites.

Reviewer #3 (Remarks to the Author):

In this manuscript, the authors reported a proof-of-concept study on the evaluation of covalent-bonded carbon environment connected to FeN₄ sites on their catalytic activity via pyrolysis-free approach. Experiments coupled with density functional theory demonstrate that the catalytic activities appear a volcano plot with the increase of degree of delocalized n electrons from the carbon matrix. This work is interesting and also provide a strategy for regulating the oxygen reduction performance of the active site by changing electronic configuration of the carbon atom covalent-connected to the FeN₄ sites. Before its publication, some major issues should be carefully addressed as below.

- 1, The Fe content in Supplementary Table S2 seems to be inconsistent with the value calculated by the illustrated structure in Figure 1.
- 2, In this work, the effect of DDE of carbon matrix is discussed, while a carbon support XC-72 was introduced in the electrochemical tests. Whether it involves in the DDE effect. What is the impact if using other support such as graphite or hard carbon (representing a high and low degree of graphitization)?
- 3, A significant variation is observed in Fe 2p XPS spectra. This is negligible because the change in the electron density of the iron center has great effects on the activity. This change can be easily detected by the isomer shift in the Fe Mössbauer spectroscopy, which is recommended.
- 4, The definition of limited current density should be described as no current platform is observed for some samples in this work.
- 5, The stability of COP should be described as FePc is known for its instability in acids. If a significant degradation can be observed in electrochemical tests, then an error bar should be exhibited in those electrochemical curves.
- 6, Tafel plots are compared neither among similar E nor similar j, which is misleading as different slopes can be observed under different overpotentials. First, the data in the larger range should be exhibited. Then, a curve of Pt/C should be added as a standard. Especially, the description of oxytropism by Tafel slopes is still controversial. The Nernstian slopes can be linked to the electron-transfer process but a higher slope is also related to other factors, which should be carefully discussed.
- 7, The TOF at 0.8 V should also be calculated for the comparison of values in literature.
- 8, Some spelling mistake should be revised, e.g. "have" is missed in page1 line 8; Figure 3d is earlier appeared than 3c.

Point-to-point response to reviewer's comments

Reviewer #1:

This paper investigated the impact of carbon moiety adjacent Fe-N_x sites in Fe-N-C catalysts. The authors combined a bottom-up synthesis with low-temperature (220 °C) pyrolysis to afford four model Fe-N-C catalysts having carbon moieties with different degree of pi conjugation. COF-Ppcfe showed the best activity, and the samples exhibited a volcano-type relationship of ORR activity and degree of pi conjugation. Some aspects of these results may be of interest in the field, and the model catalyst approach is intriguing. However, many important claims were not fully substantiated by experimental evidence. Also, the manuscript contains too many typos and awkward expression, making hard to follow the manuscript. Overall, this review cannot support the publication of this work in Nature Comm.

Response: Many thank the reviewer for providing thoughtful feedbacks. The reviewer's comments were helpful to improve the quality of our manuscript. Based on your valuable comments, we have supplied the detailed.

Comment 1: Fe contents in all catalysts should be quantified by ICP-OES analysis. Also, Mo that was used as the catalyst for generating COPs can remain after the polymerization, which should also be checked with ICP-OES analysis.

Response: We thank the reviewer's comment and suggestion. The iron content is very important. We implemented ICP-OES in the previous version and it was shown in the following Table R1, *i.e.*, **Supplementary Table S2**. In addition, we also thank the reviewers for kind suggestions. We have tested the contents of Mo element in the four samples through ICP-OES, and found that the Mo element is hardly detected.

Table R1. (*i.e.*, **Supplementary Table S2.** in the supporting information) The metal iron contents obtained by ICP-OES in four samples.

Sample	iron contents (wt.%)
COP-Ene	3.78
COP-Ppcfe	2.41
COP-Nap	1.93
COP-Pyr	1.30

Comment 2: The samples other than COP-Ppcfe should be fully characterized using STEM and EXAFS to support that all samples exclusively have Fe-N sites.

Response: We thank the reviewer's good suggestion. We have implemented STEM and EXAFS for other COPs samples and found that they do contain Fe-N bonds and we also updated the corresponding descriptions and results in the revised manuscript as follows.

“Furthermore, we performed the aberration-corrected high-angle annular dark-field scanning transmission electron microscopy (HAADF-STEM) test, which identified single Fe atoms from monodispersed bright spots images without metal clusters being observed (Fig. 2c and Supplementary Fig. 9-10) for all samples. To verify the coordination environments and chemical states of Fe atom in four samples, we carried out Fe K-edge analysis of X-ray absorption fine structure (XAFS). Synthetic four samples were demonstrated high similarity to FePc benchmark, exhibiting dominated peaks corresponding to the Fe-N (-S 1.53 Å) and Fe-N-C (-S 2.7 Å) scattering paths, whereas the Fe-Fe bond (-S 2.18 Å) was not observed (Fourier transform-extended XAFS spectra, Fig. 2d and Supplementary Fig. 12). EXAFS fitting of four samples in k-space were also consistent with that of FePc benchmark, and completely deviated from Fe foil, manifesting iron atom in synthetic samples existing as mononuclear centers (EXAFS k space fitting, Fig. 2e and Supplementary Fig. 13).”

(Page 5 in the revised manuscript)

Figure R1. (*i.e.*, **Supplementary Fig. 10.** in the revised supporting information) The HAADF STEM images of (a) COP-Ene, (b) COP-Nap and (c) COP-Pyr samples.

Figure R2. (*i.e.*, **Fig 2.** in the revised manuscript) **d**, The Fourier transforms of the EXAFS spectra for four samples, Fe foil and FePc. **e**, The corresponding EXAFS k space fitting curves of four samples, Fe foil and FePc.

Figure R3. (*i.e.*, **Supplementary Fig. 12.** in the revised supporting information) The Fourier transformations of the EXAFS spectra and corresponding EXAFS R space fitting curves of Fe foil, FePc and four synthetic samples.

Figure R4. (*i.e.*, **Supplementary Fig. 13.** in the revised supporting information) The corresponding EXAFS k space fitting curves of Fe foil, FePc and synthetic four samples.

Table R2. (*i.e.*, **Supplementary Table 1.** in the revised supporting information) EXAFS fitting parameters at the Fe K-edge various samples ($S_0=0.82$).

Sample	Path	C.N.	R (Å)	$\sigma^2 \times 10^3$ (Å ²)	ΔE (eV)	R factor
Fe foil	Fe-Fe	8*	2.47±0.01	4.5±1.2	-2.8±1.6	0.002
	Fe-Fe	6*	2.84±0.01	5.3±2.2	-5.2±3.4	
FePc	Fe-N	3.9±1.0	1.94±0.02	9.1±2.4	-0.4±3.6	0.008

	Fe-C	3.6±2.1	2.96±0.02	5.0±4.9	4.0±4.4	
COP-Ene	Fe-N	3.1±1.0	1.92±0.04	21.1±7.7	11.0±4.8	0.006
	Fe-C	5.2±1.0	3.20±0.02	5.2±3.4	7.2±1.8	
COP-Ppcfe	Fe-N	5.4±1.0	1.93±0.01	3.8±1.2	5.7±2.8	0.016
	Fe-C	7.6±2.7	2.98±0.02	2.2±2.3	9.8±3.4	
COP-Nap	Fe-N	3.8±0.5	1.98±0.02	12.0±3.7	1.2±3.1	0.014
	Fe-C	3.9±0.5	3.11±0.03	1.4±5.0	21.5±2.8	
COP-Pyr	Fe-N	3.7±0.3	1.98±0.01	8.8±1.9	3.1±1.6	0.017
	Fe-C	6.3±0.2	3.13±0.05	14.3±7.3	13.5±4.9	

^a*N*: coordination numbers; ^b*R*: bond distance; ^c σ^2 : Debye-Waller factors; ^d ΔE_0 : the inner potential correction. *R* factor: goodness of fit. * the experimental EXAFS fit of metal foil by fixing CN as the known crystallographic value.

Comment 3: Although the ORR activity of COP-Ppcfe was the best among the samples and its half-wave potential (0.748 V vs RHE) is impressive, its diffusion current density (-3 mA/cm²) is too low. According to the Levich equation with complete 4-electron selectivity, diffusion current density should be 6 mA/cm² at a rotation speed of 1600 rpm. Most of highly active Fe-N-C catalysts show diffusion current density of 5.5-6 mA/cm².

Response: We thank the reviewer for this comment. As the reviewer commented, diffusion current densities at 1600 rpm for most of high active Fe-N-C catalysts reported is 5.5-6 mA/cm². In this work, the limiting current of COP-Ppcfe is indeed not high enough as conventional Fe-N-C catalysts, which may be determined by the nature of the limited intrinsic electron conductivity in our materials and further lead to weak diffusion in oxygen reduction process. In fact, phthalocyanine materials that can stably exist in acidic media are uncommon and most reported limiting currents are generally 3-4 mA cm⁻² (*Nat. Commun.* 2013, 4, 2076; *Nat. Commun.* 2020, 11, 5283;

J. Power. Sources. 2015, 293, 511-518; *J. Mater. Chem. A*, 2019, 7, 24776-24783).

The four samples synthesized via kinetic irreversible reaction in this work show great stability in acid media and some of them exhibit four-electron pathway. Actually, this work aims at the effect of the carbon matrix on the intrinsic activity of FeN₄ moieties instead of the gap between these catalysts and the reported high-activity catalysts in this work. We believe that our multifaceted experimental and theoretical results can draw this conclusion. In future, we also believe some strategies, such as increasing initial conductivity of the materials, loading highly conductive supports, can make up for the shortcomings of our material's insufficient intrinsic conductivity and further improve the diffusion performance of these kind of materials.

Comment 4: In relation to the above question, what is four-electron selectivity of the catalysts?

Response: We thank the reviewer for this question. For the four-electron selectivity of the four catalysts, we conducted the test of the rotating ring disk electrode and calculated the electron transfer number and hydrogen peroxide yield of the four COPs and made corresponding comments in the previous version, which are displayed in **Supplementary Fig. 25.** in the supporting information.

Figure R5. (i.e., **Supplementary Fig. 25.** in the supporting information) The electrode transfer numbers and H₂O₂ yields of COP-Ene, COP-Ppcfe, COP-Nap and COP-Pyr in O₂-saturated 0.1 HClO₄ solution at a scan of 5 mV s⁻¹ and a rotation speed of 1600 rpm.

Basing on our results, the COP-Ppcfe sample is close to the four-electron reaction mechanism during the electrochemical reaction, and the H_2O_2 yield is less than 5%. For COP-Ene sample, the number of electrons transferred is close to between four electrons and two electrons, and about 10% of H_2O_2 is generated. However, the COP-Nap and COP-Pyr samples exhibited almost complete two-electron reaction under high voltage, and the H_2O_2 yield exceeded 50%.

Comment 5: Other catalysts showed very low catalytic activity? If they truly contain Fe-N sites, their activities are unusually low.

Response: We thank the reviewer' comment. As the reviewer commented, the ORR activity of other catalysts in acidic media is indeed not high enough. On the one hand, this may be caused by the slightly lower density of active sites in these catalysts. On the other hand, it may be induced by the different degree of delocalization of pai electrons in the carbon matrix adjacent to the active sites. The discrepancy in the degree of delocalization of pai electrons in the carbon matrix can induce changes of the electron configurations of the Fe-N active sites. This is also the reason that we carried out this work. Except for the MN_4 sites, the nearby carbon environment also shows nonnegligible effect on the electrocatalytic activity. In fact, similar situations also exist in pyrolytic Fe-N-C catalysts. Their structural compositions contain Fe-N bonds, but catalytic activities of some of them are not high enough as following Figure R6 (such as the ZIF-8/ Fe_{750} reference catalyst in *J. Am. Chem. Soc.* 2020, 142, 12, 5477-5481, Fig. R6a; the FeSAs/PTF-400 reference catalyst in *ACS Energy Lett.* 2018, 3, 883-889, Fig. R6b), which is more likely to be caused by discrepant carbon matrix connecting Fe-N moieties.

Figure R6. a, Electrochemical test in Ref. of *J. Am. Chem. Soc.* 2020, 142, 12, 5477-5481. b, Electrochemical test in Ref. of *ACS Energy Lett.* 2018, 3, 883-889.

Comment 6: The manuscript has too many awkward expression and typo. Some examples include

1) Abstract: Experiments coupled with density functional theory demonstrate that the catalytic activities appear a volcano plot with the increase of degree of delocalized π electrons from the carbon matrix.

2) Line 81-83: The electrochemical tests demonstrated that the oxygen reduction activities of the four COPs exhibited a volcano plot with the expansion of the carbon skeletons.

0) Line 85-89: The detailed DFT demonstrated that π -conjugated ligand configurations connected to FeN₄ sites relocate electronic filling of antibonding states in Fe atom and modulate their electronic configurations, hence altered the rate-determining step (RDS) in ORR process.

3) Title: Identify :: Identifying

4) Line 29: acutely :: acute

5) Line 34-35: “Iron and nitrogen co-doped carbons (Fe-N-C) electrocatalysts”

6) Line 54-55: “In particular, the FeN₄C₁₂, FeN₄C₈, and FeN₄C₁₀ moiety regularly separately function as typical active configuration...”

7) Line 97: “prepare-oriented”

8) Line 123: “as-synthesis”

10) Line 158: “harsh” This experimental condition is not harsh.

Response: We thank the reviewers for patiently pointing out these awkward expressions and typos in this article. We corrected the grammatical problems in the manuscript one by one. In addition, we have examined throughout the manuscript, revised them and highlighted in the yellow background in the revised version.

Some revision can be found as follows:

1) The sentence of ‘Experiments coupled with density functional theory demonstrate that the catalytic activities appear a volcano plot with the increase of degree of delocalized π electrons from the carbon matrix.’ was corrected into

‘Experiments combined with density functional theory demonstrates that the catalytic activities of these COPs materials appear a volcano plot with the increasement of delocalized π electrons in their carbon matrix.’ (Page 1 in the revised manuscript)

2) The sentence of ‘The electrochemical tests demonstrated that the oxygen reduction activities of the four COPs exhibited a volcano plot with the expansion of the carbon skeletons.’ was corrected into ‘The electrochemical tests demonstrated that the oxygen reduction activities of the four COPs exhibited a volcano plot with the expansion of the carbon skeletons adjacent to FeN₄ moieties in polymers.’ (Page 4 in the revised manuscript)

3) The sentence of ‘The detailed DFT demonstrated that π -conjugated ligand configurations connected to FeN₄ sites relocate electronic filling of antibonding states in Fe atom and modulate their electronic configurations, hence altered the rate-determining step (RDS) in ORR process’ was corrected into ‘The detailed DFT demonstrated that π -conjugated ligands connected to FeN₄ sites relocate electronic filling of antibonding states in Fe atom thereby modulating their electronic configurations and further altering their rate-determining step (RDS) in ORR process.’ (Page 4 in the revised manuscript)

4) The word of ‘Identify’ was corrected into ‘Identifying’ (Page 1 in the revised manuscript)

5) The word of ‘acutely’ was corrected into ‘acute’ (Page 2 in the revised manuscript)

1) The expression of 'Iron and nitrogen co-doped carbons (Fe-N-C) electrocatalysts' was corrected into 'Iron and nitrogen co-doped carbons (Fe-N-C)' (Page 2 in the revised manuscript)

2) The sentence of 'the $\text{FeN}_4\text{C}_{12}$, FeN_4C_8 , and $\text{FeN}_4\text{C}_{10}$ moiety regularly separately function as typical active configuration,^{14, 24-28} and thus frequently were employed as a benchmark model for active-site identification in the Mössbauer and

XANES analysis.' was corrected into 'the $\text{FeN}_4\text{C}_{12}$, FeN_4C_8 , or $\text{FeN}_4\text{C}_{10}$ moiety regularly functions as typical active configuration in different studies,^{14, 24-28} and thus frequently are employed as a benchmark model for active-site identification in the Mössbauer and XANES analysis.' (Page 2 in the revised manuscript)

6) The word of 'prepare-oriented' was corrected into 'directionally prepare' (Page 4 in the revised manuscript)

'for four samples.' (Page 5 in the revised manuscript)

3) The sentence of 'The ORR measurements for four COPs were carried out in a harsh acidic medium (0.1 M HClO_4) to evaluate their performance discrepancy' was corrected into 'The ORR measurements for four COPs were carried out in 0.1 M HClO_4 to evaluate their performance discrepancy.' (Page 6 in the revised manuscript)

4) The expressions of 'configuration', 'local environment for the FeN_x site', 'but still a great challenge', 'environment' were corrected into 'configurations'; 'local environments for the FeN_x sites'; 'but still is a great challenge'; 'environments' (Page 1 in the revised manuscript, respectively)

5) The word of 'configuration' was corrected into 'configurations' (Page 5 in the revised manuscript)

6) The expression of 'Aiming at this consistent volcano relationship plot' was corrected into 'Aiming at these consistent volcano relationship plots' (Page 7 in the revised manuscript)

14) The expression of ‘the sluggish the desorption of OH* (OH* + H⁺ + e⁻ - H₂O) into RDS’ was corrected into ‘the sluggish the desorption of OH* (OH* + H⁺ + e⁻ - H₂O) into RDS’ (Page 8 in the revised manuscript)

15) The expression of ‘further analyses that superior intrinsic active moieties’ was corrected into ‘further analyze that those superior intrinsic active moieties’ (Page 13 in the revised manuscript)

Reviewer #2:

This manuscript reports the impact of covalent-bonded carbon matrix to the so-called active site Fe-N_x by a pyrolysis-free method. Embedded Fe-N₄ into covalent organic polymer (COP) is an effective approach to keep the FeN₄ unit unchanged during the synthesis, unlike the pyrolysis process. This synthesis can investigate the configuration of FeN₄ exclusively on the ORR activity. They synthesize 4 COPs with various strengths, defined as DDEs, for the study of ORR activity and give a volcano plot. They use a series of methods to characterize the materials and also use DFT calculations as well, trying to unveil the intrinsic mechanism of ORR under the interaction between the FeN₄ and covalent-bond matrix. The manuscript is well written and the results are of interest. I would suggest its publication in Nat Comm by some revisions.

Response: We are grateful to the referee for the appreciation of our work and positive comments. We also highly appreciate your efforts in reviewing our work and giving valuable comments. Based on your valuable comments, our point-by-point responses to the comments of the referee are given as below.

Comment 1: It is better if the authors can synthesize more COPs for the volcano plot. **Response:** We thank the reviewer for the good suggestion. Firstly, we aimed at evaluating the effect of covalent-bonded carbon environment connected to FeN₄ sites on their catalytic activity by directly designing catalysts with pure carbon

environments without any N doping directly connected to the same FeN₄ sites. Therefore, we must consider that the material we design contains oxygen reduction active centers. Besides, we also need to consider the discrepancy in the conjugated systems of our designed materials. On the other hand, these designed materials must obtain good stability in acid media and exhibit certain electrocatalytic activity. We searched for many literatures to ensure that our designed materials contain both FeN₄ units and large π bonds. Eventually, we tried our best to found out four monomers containing highly symmetric cyano groups or acid anhydrides as raw materials, which could be used to synthesize COP materials. From the aspect in structural design of materials, we could design more representative structures to analyze the effect of carbon matrix on catalytic activity, such as COPs with asymmetric carbon matrix structures. However, it is still a great obstacle to experimentally achieve this type of COP material with asymmetric configurations by precise and controllable synthesis. Because, even if these COPs can be successfully synthesized, it is difficult to accurately characterize this asymmetric structure by existing technical means. In this regard, we have also carried out more theoretical designs by DFT calculations in the previous version and made further relevant discussions to convincingly evidence the conclusion in this work.

Comment 2: Besides activity, stability is also important for a catalyst. So it is better to give some investigation on this point.

Response: We thank the reviewer for this kind suggestion. We have carried out the relevant tests about stability for COP-Ppcfe sample. As shown in the following Figure R7, in the electrochemical tests after CV activation, the electrochemical curves can be kept consistent in a short time. But from a long-term current-time chronoamperometry measurement for COP-Ppcfe sample, a significant degradation indeed can be observed in electrochemical tests. Actually, the stability of the Fe-N-C catalyst in the acidic medium is related to many factors, such as demetallation, bulk carbon corrosion, surface carbon oxidation *via* Fenton reactions, and protonation of nitrogen groups followed by anion adsorption in acidic media (*Nat. Catal.*, 2020, 4,

10-19; *Energy Environ. Sci.*, 2019, 12, 3508). The stability of our pyrolysis-free COP materials may be similar to that of Fe-N-C catalysts. The synergistic effects of various factors, such as the durability of the frameworks in an electric field environment, demetalization, protonation etc., could contribute to the not high stability enough in acid ORR. Currently, there have been some reports on the preparation of catalysts with excellent stability by adopting ‘*encapsulation or confinement*’ strategies (*Adv. Mater.* 2019, 1901996; *Angew. Chem. Int. Ed.* 2013, 52, 371-375; *J. Am. Chem. Soc.* 2020, 142, 7116-7127). We believe that rational design of the COP structures through this strategy would lead to high-stability catalysts for ORR in acid media and related research work are going on in our group.

Figure R7. a, First three $j(E)$ polarization curves for the ORR on COP-Ppcfe load with XC72 disk electrode recorded at 1600 rpm in O_2 -saturated 0.1 M $HClO_4$. b, Current-time chronoamperometry for COP-Ppcfe loaded with XC72 in an O_2 -saturated 0.1 M $HClO_4$ solution at reduced potential of 0.6 V vs. RHE.

Comment 3: COPs as a model catalyst for mechanism study is fine, but the number of active sites in COPs should also be discussed as dense active sites are necessary for an efficient catalyst.

Response: We thank the reviewer for this kind suggestion. As the reviewer commented, the number of active sites does have an indispensable contribution to ORR activity. The number of active sites were determined by the Fe contents of the

four COPs by ICP-OES. In order to ensure our data more accurate and reliable, we normalized the catalytic activities by metal contents in catalysts, that is, we utilized the turnover frequency (*TOF*) and mass activity (*Ma*) per mass of metal in our previous version to study the effect of covalent-bonded carbon environment connected to FeN₄ sites on the intrinsic catalytic activity of the active sites, which avoids the effect of the number of active sites on their catalytic activity.

Comment 4: In the Introduction and Discussion, the recent reported asymmetrically electronic distribution of an active site plays important roles on the activity. This may be helpful for better understanding of the activity dependence on the FeN_x/matrix interaction through electron transfer (directly influences the electronic asymmetry). This discussion can be referred to Adv Mater 2019, 1805581; and 2020, 32 (16), 2000966.

Response: We thank the reviewer for this suggestion. According to the reviewer's suggestion, we updated the related discussions in the revised manuscripts.

“The Bader charge population analysis manifests that these carbon matrix functions as a ‘motor’, gradually donating more electrons from themselves into FeN₄ sites thereby altering the net charge of the single Fe atom (Fig. 6b), strengthening iron oxophilicity and binding with adsorbates.⁶³⁻⁶⁵” (Page 11 in the revised manuscript)

Comment 5: It should also discuss the potential design of active sites and assembly as a catalyst, which may be the topic of research for next generation catalysts. COPs or COFs (JACS 2020, 142, 8104) could be the effective approaches but require more efforts to construct catalysts with high-dense active sites.

Response: We thank the reviewers for recognition and support of COPs/COFs materials as electrocatalysts. We have discussed the related descriptions in the revised manuscripts.

“The tailorability of building blocks as well as inherent durability indwelling in covalent bonds endows them with excellent electrocatalytic characters.³⁶⁻⁴⁴ The

precise arrangement at the atomic level of their structures enables the oriented introduction of well-defined redox sites as well as electronegative heteroatoms into topological skeletons. Besides, the exclusive covalent linkages in COPs simultaneously endow them high chemical stability, which increases the feasibility of COPs application in ORR.⁴⁵” (Page 3 in the revised manuscript)

Reviewer #3:

In this manuscript, the authors reported a proof-of-concept study on the evaluation of covalent-bonded carbon environment connected to FeN₄ sites on their catalytic activity via pyrolysis-free approach. Experiments coupled with density functional theory demonstrate that the catalytic activities appear a volcano plot with the increase of degree of delocalized π electrons from the carbon matrix. This work is interesting and also provide a strategy for regulating the oxygen reduction performance of the active site by changing electronic configuration of the carbon atom covalent-connected to the FeN₄ sites. Before its publication, some major issues should be carefully addressed as below.

Response: We are grateful to the referee for the appreciation of our work and positive comments. To further improve the quality of this manuscript as well as address your concerns, we have revised our manuscript as your suggestions. In addition, we also supplied a point-by-point response as follows. We wished the revised manuscript could fulfill the requirements for the publication in Nature Communications.

Comment 1: The Fe content in Supplementary Table S2 seems to be inconsistent with the value calculated by the illustrated structure in Figure 1.

Response: We thank the reviewer’s comment. Our measured Fe contents in Supplementary Table S2 are do inconsistent with the values calculated by the

illustrated structure in Figure 1, but the overall trend of iron content in four polymers is similar as the theoretical model.

As mentioned in the comment 1 from the referee 2, we choose the kinetic irreversible covalent chemical reactions to prepare these COP materials with enough stability in acid media to obtain electrocatalytic performance without further pyrolysis in the traditional Fe-N-C case. Different from COFs/MOFs, the as-synthesized COP can only show the well-defined molecular structure in short range and cannot inevitably keep the crystalline framework in the long range due to the kinetic irreversible covalent chemistry. Additionally, since the polymerization reaction in this work is a solid-phase reaction, all iron atoms are hard to be riveted in the polymer skeleton materials. Therefore, the actual iron contents obtained are inferior to the theoretical molecular models in our work.

As the response to the comment 3 from the referee 2, in order to ensure our data more accurate and reliable, we normalized the catalytic activities by metal contents in catalysts, that is, we utilized the turnover frequency (*TOF*) and mass activity (*Ma*) per mass of metal to study the effect of covalent-bonded carbon environment connected to FeN₄ sites on the intrinsic catalytic activity of the active sites, which avoids the effect of the number of active sites on their catalytic activity.

Table R3. The metal iron contents obtained by ICP-OES in four samples, *TOF* value and *Ma* per mass of metal at 0.7 V versus RHE.

Sample	iron contents (wt.%)	TOF @ 0.7 V (Hz)	Ma (Ma mg⁻¹_{metal})
COP-Ene	3.78	0.0019	13.49
COP-Ppcfe	2.41	0.13	896.59
COP-Nap	1.93	0.00118	8.13
COP-Pyr	1.30	0.000875	6

Comment 2: In this work, the effect of DDE of carbon matrix is discussed, while a carbon support XC-72 was introduced in the electrochemical tests. Whether it involves in the DDE effect. What is the impact if using other support such as graphite or hard carbon (representing a high and low degree of graphitization)?

Response: We thank the reviewer's good comment and suggestion. The carbon support XC-72 introduced in the electrochemical tests may involve in a slight DDE effect, but we have minimized the effect when designing the experiment. First, we did an acidic ORR test experiment with pure COP materials without foreign carbon supports, and made corresponding discussions in the previous version (**Fig. S19**). It can be observed that although the ORR performance of the four COP materials in acidic media is slightly low, their onset potential still satisfies the volcano plot relationship. Second, considering the limited conductivity of these materials, we only added the carbon support XC-72 in the electrochemical test instead of during the experimental synthesis, and the same amount of carbon support is added under the same operating conditions, which aims to increase the conductivity of COP materials while minimizing the interactions between the carbon support and the COPs, that is, eliminating the DDE effect of carbon support.

Figure R8. (*i.e.*, **Supplementary Fig. 19** in the supporting information) Electrochemical characterization. (a) LSV curves (b) onset potential of COP-Ene, COP-Ppcfe, COP-Nap and COP-Pyr without Cabot Vulcan XC-72 in O₂-saturated 0.1 HClO₄ solution at a scan of 5 mV s⁻¹ and a rotation speed of 1600 rpm.

'To exclude the influence of the ORR activity of the conductive agent XC72, we tested their ORR activity without the conductive agent in 0.1M HClO₄. The activity trend (initial potential) of the different samples remained consistent with the condition of carbon black.' (The descriptions in supporting information)

At the same time, we also approve of reviewers' good suggestion very much. In addition to hard carbon (XC-72, typical hard carbon) loaded COP materials, we added soft carbon (graphite) and rigid carbon (carbon nanotube, CNT) as supports, and can draw the conclusion: regardless of the type of carbon support supported by these polymers, their catalytic activity of these polymers still exhibits a volcano relationship.

Figure R9. (i.e., Supplementary Fig. 22. in the revised supporting information) a, LSV curves of COP-Ene, COP-Ppcfe, COP-Nap, and COP-Pyr samples loaded with XC-72. in O₂-saturated 0.1 HClO₄ solution. b, LSV curves of COP-Ene, COP-Ppcfe, COP-Nap, and COP-Pyr samples loaded with graphite in O₂-saturated 0.1 HClO₄ solution. c, LSV curves of COP-Ene, COP-Ppcfe, COP-Nap, and COP-Pyr samples loaded with CNT in O₂-saturated 0.1 HClO₄ solution.

Comment 3: A significant variation is observed in Fe 2p XPS spectra. This is non-negligible because the change in the electron density of the iron center has great effects on the activity. This change can be easily detected by the isomer shift in the Fe Mössbauer spectroscopy, which is recommended.

Response: We thank the reviewer's good comment and suggestion to improve the quality of our manuscript. We have carried out Mössbauer spectroscopy for four COPs samples and updated the corresponding descriptions and results in the revised manuscript as follows.

Figure R10. (*i.e.*, Fig. 5. in the revised manuscripts) **The effect of DDE in carbon matrix adjacent to FeN₄ moieties on Fe electronic configurations from fitted Mössbauer spectra.** **a-d**, Room-temperature ⁵⁷Fe Mossbauer spectra of COP-Ene, COP-Ppcfe, COP-Nap and COP-Pyr samples, respectively. **e**, The corresponding isomer shift (IS) of D1 and D3 species in four synthetic COPs from ⁵⁷Fe Mössbauer measurements. **f**, Contents of different Fe moieties (D1, D2, D3) in four synthetic COPs.

Table R4. (*i.e.*, Supplementary Table 3. in the revised Supporting Information)

Parameters derived from the fittings of Mössbauer spectra.

	Component	IS mm s ⁻¹	QS mm s ⁻¹	LW mm s ⁻¹	Area %	Assignment
COP-Ene	D3	0.281	0.934	0.339	14.05	HS FeN ₄
	D1	0.173	0.517	0.554	85.95	LS FeN ₄
COP-Ppcfe	D3	0.12203	1.81541	0.43278	33.95	HS FeN ₄
	D1	0.24914	0.78036	0.83689	28.73	LS FeN ₄
	D2	0.18927	2.8433	0.55351	37.32	HS FeN ₄
COP-Nap	D3	0.28	2	0.682	21.87	HS Fe N ₄
	D1	0.29	0.57429	0.682	78.13	LS FeN ₄
COP-Pyr	D3	0.3	1.75037	0.594	16.67	HS FeN ₄
	D1	0.33205	0.56294	0.60807	83.33	LS FeN ₄

“From the Fe 2p XPS spectra of COP-Ene, COP-Ppcfe, COP-Nap and COP-Pyr, we have observed that the Fe moieties in synthetic four COP samples could be deconvoluted into Fe $2p_{1/2}$ species and Fe $2p_{3/2}$ species, which suggested that Fe moieties in different COPs contain simultaneously electronic states of Fe²⁺ and Fe³⁺ (Supplementary Fig. 15). Interestingly, the binding energy of Fe $2p_{1/2}$ and Fe $2p_{3/2}$ species in four samples shift closer to the higher position with the increasement of DDE, respectively, meaning that the bonding strengths of covalent bonds in the as-prepared COPs are gradually strengthening.³⁸ In order to probing the variations in the electron density of the iron center more accurately, we performed room-temperature ⁵⁷Fe Mössbauer spectra for these synthesized samples. As shown in Fig.5a-d, the Mössbauer spectra could be well fitted by two or three doublets, which ascribed to low-spin FeN₄ (D1, green doublets), high-spin FeN₄ (D2, brown doublets), and high-spin FeN₄ (D3, blue doublets), respectively. No signals of iron carbide, iron oxide, and iron were detected, suggesting that these as-synthesized COPs contains exclusively isolated single iron sites with defined N coordination. For the D1 species, the isomer shift (IS) values are gradually aggrandizing with the amplification of DDE

in the carbon matrix adjacent to FeN₄ moieties (Fig. 5e), which indicates that electron densities of the iron centers in s orbits of D1 species are gradually decreasing with the increasement of DDE in carbon matrix adjacent to FeN₄ moieties. For the D3 species, the IS values exhibit an inverted volcano plot with the increasement of DDE (Fig. 5e), which suggests that electron densities of the iron centers in s orbits of D3 species is increasing first and then decreasing with the increasement of DDE in carbon matrix. Intriguingly, the tendency of electron densities of the iron centers in s orbit of D3 species is exactly consistent to the catalyst-activity volcano plot, illustrating that the D3 species significantly promote the acidic ORR process.^{13, 15, 63} In addition, the appearance of neo-generated species, i.e. D2 species in COP-Ppcfe sample (Fig. 5b, 5f), reveal that a proper DDE in carbon matrix might promote generation of fresh derivatives stemmed from electronic state of Fe atom.^{14, 64}” (Page 10 in the revised manuscript)

Comment 4: The definition of limited current density should be described as no current platform is observed for some samples in this work.

Response: We thank the reviewer's kind comment and suggestion. We add the following description in the original text:

“As to the LSV of COP-Nap and COP-Pyr without platform, the current density corresponding to 0 potential on the linear sweep volt-ampere curve was considered as the current limiting density” (Electrochemical Measurements section in the Supplemental Information)

Comment 5: The stability of COP should be described as “FePc is known for its instability in acids. If a significant degradation can be observed in electrochemical tests, then an error bar should be exhibited in those electrochemical curves.

Response: We thank the reviewer's kind comment and suggestion. The stability of iron phthalocyanine (FePc) under acidic conditions is indeed very unstable, but for their polymers, their stability is slightly improved (*Nanoscale*, 2015, 7, 11633-11641). Therefore, in the electrochemical test after CV activation, the electrochemical curve

can be kept consistent in a short time. As mentioned as the comment 2 from the referee 2, but from a long-term current-time chronoamperometry measurement for COP-Ppcfe sample, a significant degradation indeed can be observed in electrochemical tests, therefore, we supplemented error bar in the electrochemical curve.

Figure R11. a, First three $j(E)$ polarization curves for the ORR on COP-Ppcfe load with XC72 disk electrode recorded at 1600 rpm in O_2 -saturated 0.1 M $HClO_4$. b, Current-time chronoamperometry for COP-Ppcfe loaded with XC72 in an O_2 -saturated 0.1 M $HClO_4$ solution at reduced potential of 0.6 V vs. RHE.

Comment 6: Tafel plots are compared neither among similar E nor similar j, which is misleading as different slopes can be observed under different overpotentials. First, the data in the larger range should be exhibited. Then, a curve of Pt/C should be added as a standard. Especially, the description of oxytropism by Tafel slopes is still controversial. The Nernstian slopes can be linked to the electron-transfer process but a higher slope is also related to other factors, which should be carefully discussed.

Response: We thank the reviewer's kind comment and suggestion.

1) According to the reviewer's suggestion, we first corrected Figure 3c. We have adjusted the comparisons of Tafel curves to a similar J, and put these data into a larger range for evaluation. At the same time, we have also supplied the Tafel curve of Pt/C for comparison.

Figure R12. (i.e., Fig. 3c in the revised manuscript) Tafel plots of four samples and Pt/C.

2) We thank the reviewer's kind comment and suggestion again. We removed the description of oxytropism by Tafel slopes and amended it in the original text as follows.

'the changes of Tafel slopes also reflect the tendency of kinetic processes in ORR to become faster first and then slower with the increasement of DDE in carbon matrix adjacent to FeN₄ moieties.' (Page 7 in the revised manuscript)

Comment 7: The TOF at 0.8 V should also be calculated for the comparison of values in literature.

Response: Thank you for the reviewer's kind suggestion. We have supplemented the TOF at 0.8 V for four COPs in the revised supporting information.

Figure R13 (*i.e.*, **Supplementary Fig. 21.** in the revised supporting information). (a) The kinetic current density (J_k) @ 0.8 V of COP-Ene, COP-Ppcfe, COP-Nap and COP-Pyr. (b) The turnover frequency at 0.8 V of COP-Ene, COP-Ppcfe, COP-Nap and COP-Pyr.

Comment 8: Some spelling mistake should be revised, e.g. “have” is missed in page1 line 8; Figure 3d is earlier appeared than 3c.

Response: We thank the reviewer's kind comment and suggestion. We have carefully checked our manuscript, corrected the typos, and adjusted the order of describing the contents of Figure 3c and Figure 3d in this revised version.

REVIEWER COMMENTS

Reviewer #1 (Remarks to the Author):

The authors have made a substantial revision to the original manuscript. The revised manuscript now appears suitable for publication in Nature Comm.

Reviewer #2 (Remarks to the Author):

The response is fine with me. The authors have addressed all of my concerns so its publication is recommended.

Reviewer #3 (Remarks to the Author):

After the revision, most of my doubts are solved. However, a key issue is still remained because of "the precise arrangement of the active site at atomic level" stated in this manuscript seems to be failed according to Mossbauer spectrums.

For the fundamental study of Fe-N-C, a critical challenge is to synthesize the active site with a clear structure. Though the pyrolysis routine is inherent deficiency as described by the authors, but the complex state of the active site is also observed in this work prepared by a pyrolysis-free method as the observed two or three doublets are similar to the traditional Fe-N-C prepared by pyrolysis. But in coordination compound such as FePc, only one doublet is generally observed (e. g. 10.1038/s41467020-18062-y).

Moreover, the assignment of those doublets is somewhat hasty: first, the number of significant digits is disordered from 1 to 5 (2, 0.3, 0.28, 0.682, 0.12203); then the parameters are entirely groundless, which can be observed as follows:

Sample IS QS

D1 COP-Ene 0.173 0.517

COP-Ppcfe 0.24914 0.78036

COP-Nap 0.29 0.57429

COP-Pyr 0.33205 0.56294

Ref 15 0.33±0.02 1.0±0.05

Ref 17 0.34 0.94

D3 COP-Ene 0.281 0.934

COP-Ppcfe 0.12203 1.81541

COP-Nap 0.28 2

COP-Pyr 0.3 1.75037

From the table, the assignment of Mossbauer spectrums in the submitted work is quite ambiguous. Neither the IS nor the QS is related in one group of doublets. For example, the IS and QS of D3 in COP-Ene is somewhat similar to that of D1 in traditional pyrolyzed Fe-N-C. The IS and QS of D3 in COP-Nap, however, is somewhat similar to that of D2 in traditional pyrolyzed Fe-N-C. Therefore, the explain of the spectrum is misleading.

To improve the manuscript, two suggestions are given:

1. The reason for the two or three doublets observed in one sample should be explained, now that only one kind of active site exists in each sample.
2. The explanation of the Mossbauer spectrums should be more careful. Especially, now that the structure of the active site is clear in this work, the IS and QS can be directly calculated according to the structure by a DFT-based method developed recently (10.1021/acscatal.9b02586). Then the theoretical and experimental values should be compared.

Point-to-point response to reviewer's comments

Reviewer #1

The authors have made a substantial revision to the original manuscript. The revised manuscript now appears suitable for publication in Nature Comm.

Response: Thank you reviewer for your comment.

Reviewer #2

The response is fine with me. The authors have addressed all of my concerns so its publication is recommended.

Response: Thank you reviewer for your comment.

Reviewer #3

Comment: After the revision, most of my doubts are solved. However, a key issue is still remained because of “the precise arrangement of the active site at atomic level” stated in this manuscript seems to be failed according to Mössbauer spectrums. For the fundamental study of Fe-N-C, a critical challenge is to synthesize the active site with a clear structure. Though the pyrolysis routine is inherent deficiency as described by the authors, but the complex state of the active site is also observed in this work prepared by a pyrolysis-free method as the observed two or three doublets are similar to the traditional Fe-N-C prepared by pyrolysis. But in coordination compound such as FePc, only one doublet is generally observed (e. g. 10.1038/s41467-020-18062-y). Moreover, the assignment of those doublets is somewhat hasty: first, the number of significant digits is disordered from 1 to 5 (2, 0.3, 0.28, 0.682, 0.12203); then the parameters are entirely groundless, which can be observed as follows:

Sample IS QS:

D1 COP-Ene 0.173 0.517;
COP-Ppcfe 0.24914 0.78036;
COP-Nap 0.29 0.57429;
COP-Pyr 0.33205 0.56294;
Ref 15 0.33±0.02 1.0±0.05;
Ref 17 0.34 0.94;
D3 COP-Ene 0.281 0.934;
COP-Ppcfe 0.12203 1.81541;
COP-Nap 0.28 2;
COP-Pyr 0.3 1.75037.

From the table, the assignment of Mössbauer spectrums in the submitted work is quite ambiguous. Neither the IS nor the QS is related in one group of doublets. For example, the IS and QS of D3 in COP-Ene is somewhat similar to that of D1 in

traditional pyrolyzed Fe-N-C. The IS and QS of D3 in COP-Nap, however, is somewhat similar to that of D2 in traditional pyrolyzed Fe-N-C. Therefore, the explain of the spectrum is misleading. To improve the manuscript, two suggestions are given:

Response: Thanks for your comments. After seriously considering the comments, we re-tested our four samples and the benchmark FePc (directly bought from *Macklin Biochemical Co., Ltd.*) and fitted Mössbauer curves. We keep two significant digits after the decimal point for all parameters of the fitting for Mössbauer spectrums in this response. The test and analysis results are shown as follows (*Fig. R1 and Table R1*). From this second test results, we unexpectedly found that the benchmark FePc did not fit only one doublet like the reference given by the reviewer. Our test results about FePc can also be fitted to three doublets. The relevant literature shows that the doublet A2 can be attributed to the FeN₄ site and the doublet A1 and A3 can be attributed to iron microenvironments absorbed oxygen between the layers of iron phthalocyanines (*J. Phys. Chem. 1980, 84, 1936-1939; Hyperfine Interactions 139/140: 631-639, 2002.*). The difference between the doublet A1 and A3 may be due to the molecular stacking arrangements between FePc molecules, resulting in small differences in the interaction with oxygen (*Fig. R2*). This result is unexpected for us, which may be attributed to the extremely high sensitivity to energy changes on the order of 10⁻⁸ eV (ca. 10⁻⁴ cm⁻¹) and extreme sharpness of tuning (ca. 10⁻¹³) for Mössbauer spectroscopy (*page 2 in the book 'Mössbauer Spectroscopy and Transition Metal Chemistry'*). For a sample with the single pure component, a slight structural change in the sample tissue will be sensitively detected by Mössbauer spectroscopy. FeN₄ moieties is generally recognized to be very easy to adsorb oxygen, two or three doublets detected by the Mössbauer spectroscopy may be inevitable. As for the FePc in ref *10.1038/s41467-020-18062-y*, although only one doublet is generally observed, firstly, the experimental data and related parameters (IS, QS, line width, etc.) about this doublet is unclear. Then, this doublet exists in the form of asymmetric peaks, however, the fitting doublets of the Fe-N-C sites in the pyrolysis catalyst mostly exists in the form of symmetrical doublets. From this point, the FeN₄ structure between natural FePc and phthalocyanine type FeN₄ structure in the pyrolysis catalyst at least is still very different (*vide infra*).

Because building blocks in our COP-Ppcfe sample extremely resemble that of the benchmark FePc, the test results about COP-Ppcfe sample are also greatly consistent with unit of the FePc. Therefore, the doublets may be assigned as: the doublet C2 to FeN₄ site, the doublet C1 and doublet C3 to a small amount of iron microenvironments containing oxygen between the layers of iron phthalocyanines. The difference between the doublet C1 and C3 may be due to the staggered layers in COP-Ppcfe resulting in a slight difference with adsorbed oxygen. As for COP-Nap and COP-Pyr samples, from the molecular structure (pyridine N connected to Fe instead of pyrrole N) and ORR test results (similar LSV curves), we can predict that their active sites should be particularly similar in theory. In our latest test results, the IS and QS values are similar, which is consistent with theory. Combined with result of

XAFS, we attribute main doublet D1 and doublet E1 to FeN₄ sites. For doublet D2 and doublet E2, because of their QS values similar to doublet A3 sites in FePc, they also may be attributed to iron microenvironments containing absorbed oxygen between the layers of iron phthalocyanines. Similarly, for COP-Ene, doublet B1 may be attributed to FeN₄ site, and doublet B2 may be attributed to FeN₄ moiety absorbed oxygen.

Generally speaking, as for pyrolytic FeN₄ catalysts, the reported references suggest that the IS value is nondiscriminating whereas the QS value reflects the structural changes of the active sites. All pyrolytic Fe-N-C catalysts comprising FeN_xC_y moieties have shown at least two distinct doublets in their Mössbauer spectra, often labeled D1 and D2. Their reported values of QS at room temperature are in the range 0.90-1.25 and 2.0-2.8 mm·s⁻¹, respectively (*ACS Catal.* 2019, 9, 9359-9371). In our samples, the changes about IS (the maximum value of 0.33 mm s⁻¹ in E1, the minimum value of 0.03 mm s⁻¹ in A3) are also nondiscriminating. For the deviation about QS between the doublet B2 site (QS=1.09) in COP-Ene and the doublet D2 site (QS=1.74) in COP-Nap, this change may be caused by the changes in the degree of delocalization of the benzene ring adjacent to FeN₄ sites, which is similar to the case of FeN₄C₈ and FeN₄C₁₀ moieties in pyrolyzed Fe-N-C catalyst (*Energy Environ. Sci.* 2019, 12, 2548-2558). Since our samples do not have relevant reference materials for structural comparison and the samples we prepared are different from traditional pyrolysis catalysts in many aspects, such as molecular stacking arrangements, Fe local surrounding environment, carbon matrix type, microcrystalline structure, it may be irrational to compare the IS and QS values of our pyrolysis-free samples with that of the pyrolyzed Fe-N-C catalysts to some extent.

Actually, as mentioned in our manuscript, our XAFS can confirm at the atomic level that our pyrolysis-free samples are dominated by FeN₄ active sites. Supplemented by STEM, NMR spectra, XPS, FT-IR, we can prove that the samples we synthesized are almost consistent with our proposed models. DFT calculations, ORR testing combined some experimental results confirmed the degree of carbon delocalization in carbon matrix affects ORR performance. More convincingly, we later found that Mukerjee et al. have ever reported that delocalized π -electrons optimize oxygen-reduction activity through different experimental methods and ideas (*J. Am. Chem. Soc.* 2013, 135, 15443-15449). Therefore, our conclusion in this manuscript is convincing and the methods of argument is multi-scale and novel. Although the Mössbauer spectrums have 2-3 peaks in our samples, fitting results of XAFS manifest that most of the active sites in our sample still exist in a single FeN₄ structure. Besides, oxygen adsorption is the necessary first step of the ORR process (refer to DFT calculations), so we believe that a small amount of oxygen adsorption here will not have a particularly large impact on the experimental results. Although in the detection about Mössbauer spectrums, our method may does show slight complexities, it still has great advantages, such as structural adjustability, tailorability of building block, which is difficult to substituted by pyrolysis preparation. In addition, it is very common to detect five to six sets of doublets for traditional pyrolysis catalysts (*Phys. Chem. Chem. Phys.*, 2012, 14, 11673-11688; *J. Am. Chem.*

Soc. 2014, 136, 978-985). Therefore, in terms of the complexity of the active site, our pyrolysis-free method has its advantage compared with the general pyrolysis method in terms of the specificity of the active site.

Although the reviewer pointed out in the first revision, *i.e.* “significant variation in Fe 2p XPS spectra can be easily detected by the isomer shift in the Fe Mössbauer spectroscopy”, it may be also irrational to use the Fe Mössbauer spectroscopy in this work basing on the above investigations. XPS characterization focuses more on the changes of the electronic state in a certain metal orbital (such as Fe 2p 1/2, Fe 2p 3/2), while the IS value in the Mössbauer spectrums reflects the difference in the total density about all s-electrons at the absorbing nucleus of Fe atoms. The comparison from the XPS to the IS value in the Mössbauer spectrums has a large inconsistency regardless of in both the spatial scale and the time scale. In order to make the content of this work more rigorous, we have removed the relevant descriptions about Mössbauer spectrums and revised the statements about “the precise arrangement of the active site at atomic level” in the updated manuscript as follows,

“Their well-defined configuration enables the oriented introduction of redox sites as well as electronegative heteroatoms into topological skeletons.” (Page 3 in the revised manuscript)

Figure R1. Room-temperature ^{57}Fe Mössbauer spectra of FePc, COP-Ene, COP-Ppcfe, COP-Nap and COP-Pyr samples, respectively.

Table R1. Parameters derived from the fittings of Mössbauer spectra.

	Component	IS mm s ⁻¹	QS mm s ⁻¹	LW mm s ⁻¹	Area %	Assignment
FePc	A1	0.19	0.42	0.46	15.33%	MS or LS FeN ₄ -O ₂
	A2	0.18	2.72	0.29	69.43%	FeN ₄
	A3	0.03	1.55	0.58	15.24%	HS FeN ₄ -O ₂
COP-Ene	B1	0.18	0.54	0.56	92.76%	FeN ₄
	B2	0.33	1.09	0.27	7.24%	HS FeN ₄ -O ₂
COP-Ppcfe	C1	0.26	0.79	0.58	20.99%	MS or LS FeN ₄ -O ₂

	C2	0.18	2.89	0.51	61.85%	FeN ₄
	C3	0.09	1.89	0.57	17.16%	HS FeN ₄ -O ₂
COP-Nap	D1	0.29	0.55	0.58	74.42%	FeN ₄
	D2	0.26	1.74	0.59	25.58%	HS FeN ₄ -O ₂
COP-Pyr	E1	0.33	0.56	0.58	81.94%	FeN ₄
	E2	0.30	1.74	0.59	18.06%	HS FeN ₄ -O ₂

Figure 1. Structure of α - and β -polymorphs of iron phthalocyanines.

Figure R2. a, Structure of α - and β -polymorphs of iron phthalocyanines; b, Room-temperature ^{57}Fe Mössbauer spectra of FePc (pure) samples (ref. The Journal of Physical Chemistry, Vol. 84, No. 15, 1980).

1. The reason for the two or three doublets observed in one sample should be explained, now that only one kind of active site exists in each sample.

Response: Thanks for your comments. As mentioned above, the reason for two or three doublets in one sample may be that in addition to their inherent FeN₄ sites in each sample, the sample may inevitably contain FeN₄ sites adsorbed a small amount of oxygen from the air. Owing to the extremely high sensitivity to energy changes on the order of 10^{-8} eV (ca. 10^{-4} cm⁻¹) and extreme sharpness of tuning (ca. 10^{-13}) for Mössbauer spectroscopy, a slight structural change for a sample with the single pure component will be sensitively detected by Mössbauer spectroscopy. FeN₄ moieties are generally recognized to be very easy to adsorb oxygen, FeN₄ moieties with oxygen detected by the Mössbauer spectroscopy may be inevitable. Nonetheless, our fitting results of XAFS manifest that active sites in our sample mainly still exist in a single FeN₄ structure. Besides, oxygen adsorption is the necessary first step of the ORR process (refer to DFT calculations), so we believe that a small amount of oxygen adsorption here will not have a particularly large impact on the experimental results.

2. The explanation of the Mössbauer spectrums should be more careful. Especially, now that the structure of the active site is clear in this work, the IS and QS can be directly calculated according to the structure by a DFT-based method developed recently (10.1021/acscatal.9b02586). Then the theoretical and experimental values should be compared.

Response: Thank you for your comments. As mentioned as above, we re-tested our four samples and the benchmark FePc, fitted and re-analyzed Mössbauer curves. Due to the limitation of experimental conditions, it is hard to carry out the related DFT-based method about IS and QS values. The calculations involve various of choices about basis sets and functionals, it is necessary to combine numerous experimental studies to draw relatively accurate conclusions. So, it may be difficult to guarantee experimental and theoretical calculations to achieve a good match. Besides, since IS does not change much, the results about a change trend of QS rather than a specific value of QS may be obtained (similar to the DFT calculations in electrochemical testing), which may be lack of significance for this work. Importantly, a very objective status quo is that our laboratory does not have the relevant basis for the calculations of the Mössbauer parameters. In addition, in the prevalence of the current global epidemic about the Covid-19 pandemic, achieving the calculation of the Mössbauer parameters is extremely difficult to us in a short period. In order to make the content of this work more rigorous, we have removed the relevant descriptions about Mössbauer spectrums.

REVIEWER COMMENTS

Reviewer #3 (Remarks to the Author):

All the questions proposed by the reviewers are well answered by the authors. The revised manuscript is acceptable to be published on Nature communications.

Point-to-point response

Reviewer #3

Comments: All the questions proposed by the reviewers are well answered by the authors. The revised manuscript is acceptable to be published on Nature communications.

Response: Thank you for your approval